# Phosphoglycerate dehydrogenase activates PKM2 to phosphorylate histone H3T11 and attenuate cellular senescence

Yinsheng Wu [1,6], Lixu Tang[2,6], Han Huang[1], Qi Yu [1], Bicheng Hu[3], Gang Wang[1], Feng Ge [4], Tailang Yin [5] ✉, Shanshan Li [1] ✉ & Xilan Yu [1] ✉

Vascular endothelial cells (ECs) senescence correlates with the increase of cardiovascular diseases in ageing population. Although ECs rely on glycolysis for energy production, little is known about the role of glycolysis in ECs senescence. Here, we report a critical role for glycolysis-derived serine biosynthesis in preventing ECs senescence. During senescence, the expression of serine biosynthetic enzyme PHGDH is significantly reduced due to decreased transcription of the activating transcription factor *ATF4*, which leads to reduction of intracellular serine. PHGDH prevents premature senescence primarily by enhancing the stability and activity of pyruvate kinase M2 (PKM2). Mechanistically, PHGDH interacts with PKM2, which prevents PCAF-catalyzed PKM2 K305 acetylation and subsequent degradation by autophagy. In addition, PHGDH facilitates p300-catalyzed PKM2 K433 acetylation, which promotes PKM2 nuclear translocation and stimulates its activity to phosphorylate H3T11 and regulate the transcription of senescence-associated genes. Vascular endothelium-targeted expression of PHGDH and PKM2 ameliorates ageing in mice. Our findings reveal that enhancing serine biosynthesis could become a therapy to promote healthy ageing.

Ageing is a major risk factor for most human diseases, including cancer, heart diseases and diabetes[1]. One hallmark of ageing is cellular senescence, which is a permanent form of cell cycle arrest triggered by accumulated stochastic damage to cells[2]. Clearance of naturally occurring senescent cells can ameliorate the age-related pathologies and extend lifespan and healthspan in mouse model[3]. Vascular endothelial cells (ECs) form the inner lining of blood vessels and the senescent ECs are proposed to contribute to vascular ageing and age-related vascular diseases, particularly atherosclerosis and hypertension[4,5]. Cellular senescence is regulated through a variety of networks triggered by extracellular and intracellular stimuli[6]. Elucidating the senescence regulatory mechanism could provide potential intervention targets to attenuate degenerative processes and improve the life quality of the ageing population[6].

Alterations in energy metabolism is an important hallmark of ageing and metabolic interventions can delay ageing[7,8]. Mitochondrial dysfunction, disrupted $NAD^+$ metabolism and hyperglycemia have been shown as major metabolic drivers of senescence[9]. Methionine restriction has been shown to slow down the senescence of human fibroblasts and extend lifespan in animal models[10,11]. Some metabolic enzymes have been reported to regulate cellular senescence. For example, pyruvate dehydrogenase (PDH) is activated to mediate oncogene-induced senescence and associated metabolic rewiring[12]. One important mechanism by which cell metabolism regulates

[1]State Key Laboratory of Biocatalysis and Enzyme Engineering, School of Life Sciences, Hubei University, Wuhan, Hubei 430062, China. [2]School of Martial Arts, Wuhan Sports University, Wuhan, Hubei 430079, China. [3]The Central Laboratory, Wuhan No.1 Hospital, Wuhan, Hubei 430022, China. [4]Key Laboratory of Algal Biology, Institute of Hydrobiology, Chinese Academy of Sciences, Wuhan, Hubei 430072, China. [5]Reproductive Medicine Center, Renmin Hospital of Wuhan University, Wuhan, Hubei 430060, China. [6]These authors contributed equally: Yinsheng Wu, Lixu Tang. ✉e-mail: reproductive@whu.edu.cn; shl@hubu.edu.cn; yuxilan@hubu.edu.cn

senescence is altering epigenetic modifications, including histone modifications, DNA methylation, etc.[13]. We have previously reported that acetate accelerates premature senescence of endothelial cells (ECs) by increasing intracellular acetyl-CoA synthesis, inducing histone H4K16 acetylation (H4K16ac) and causing telomere shortening[14].

Glucose metabolism is emerging as a key regulator of endothelial cells. For example, the proliferating endothelial cells rely on glycolysis to generate ~85% ATP, which is similar to the "Warburg effect" in cancer cells but different from other normal cell types[15,16]. The pyruvate kinase M2 (PKM2) is a predominant form of pyruvate kinase expressed in endothelial cells and is required for ECs growth[17,18]. Recent study showed that glycolysis-derived L-serine (hereinafter referred to as serine) biosynthesis is required for survival of endothelial cells by promoting heme synthesis[19]. The glycolytic intermediate 3-phosphoglycerate (3-PG) fuels synthesis of non-essential amino acid serine via a three-step reaction involving three metabolic enzymes: phosphoglycerate dehydrogenase (PHGDH), phosphoserine amino-transferase 1 (PSAT1) and phosphoserine phosphatase (PSPH) (Fig. 1a). Loss of PHGDH leads to reduced intracellular serine biosynthesis and causes growth defect of ECs; however, exogenous added serine cannot rescue the growth defects caused by deficiency of PHGDH[19], suggesting that PHGDH may have other functions in ECs biology. Most importantly, it remains unknown about the functions of glycolytic enzymes and metabolites in ECs senescence.

In this work, we compared the transcriptomes of young and senescent ECs and found the glycolysis-derived de novo serine synthesis pathway is significantly reduced, especially for PHGDH. Further study showed that PHGDH plays both serine-dependent and serine-independent functions in mitigating ECs senescence. PHGDH promotes the stability and activity of PKM2 to phosphorylate histone H3T11, which constitutes the PHGDH-PKM2-H3pT11 axis to regulate the expression of senescence-associated genes and prevent premature senescence. Most importantly, vascular endothelium-targeted expression of PHGDH and PKM2 ameliorates the ageing phenotype and improves the cardiac functions in mice. Together, these results not only reveal a previously unknown mechanism by which PHGDH/PKM2 regulates cellular senescence but also suggest enhancing serine bio-synthesis as a potential anti-ageing therapy.

## Results

### PHGDH-mediated serine biosynthesis is decreased in senescent endothelial cells

To identify genes differentially expressed during cellular senescence, we performed RNA sequencing (RNA-seq) for young (P6) and replicative senescent (P20) human umbilical vein endothelial cells (HUVECs) that continued passaging from the same umbilical cord (Supplementary Fig. 1a). Overall, a total of 4037 genes were differentially expressed during senescence, of which 1707 genes were upregulated ($\log_2$(FC)≥0.75, $P$<0.05) and 2330 genes were downregulated ($\log_2$(FC)≤−0.75, $P$<0.05) in senescent HUVECs (Supplementary Fig. 1b). Kyoto Encyclopedia of Genes and Genomes (KEGG) pathway analysis showed that these differentially expressed genes were enriched in cell cycle, cellular senescence, p53 signaling pathway, glycine, serine and threonine metabolism etc. (Supplementary Fig. 1c). Gene set enrichment analysis (GSEA) also revealed that genes involved in glycine, serine and threonine metabolism were significantly regulated during cellular senescence, including *PHGDH*, *PSAT1*, *PSPH* and *SHMT1/2* (Fig. 1b), among which PHGDH, PSAT1 and PSPH catalyze glycolysis-derived de novo serine synthesis (Fig. 1a)[20]. By quantitative reverse transcription PCR (RT-qPCR), we confirmed that these genes, especially *PHGDH* were remarkably downregulated in replicative senescent HUVECs (Fig. 1c). We also observed a corresponding decrease in the protein levels of these enzymes in replicative senescent HUVECs (Fig. 1d). In addition, the reduction of PHGDH was observed in hydrogen peroxide ($H_2O_2$)-induced senescent HUVECs and DNA damaging agent etoposide-induced senescent HUVECs (Supplementary Fig. 1d, e).

During replicative senescence, the largest reduced serine bio-synthetic enzyme is PHGDH (Fig. 1c, d), which catalyzes the committed step in de novo serine synthesis in ECs[19]. Two independent small interfering RNAs (siPHGDH) were used to knock down the expression of PHGDH in HUVECs. The intracellular serine was significantly reduced in siPHGDH cells (Supplementary Fig. 1f). Consistent with reduced expression of PHGDH, the intracellular serine was significantly decreased in senescent HUVECs (Fig. 1e).

The transcription of *PHGDH* has been reported to be induced by the activating transcription factor 4 (ATF4)[21,22]. ATF4 is a member of the ATF/CREB family of transcription factors and plays pivotal roles in physiological responses to stresses including hypoxia, endoplasmic reticulum stress, amino acid deprivation, oxidation, and mitochondrial stress[23]. The mRNA and protein levels of ATF4 were significantly reduced during senescence (Fig. 1f). Knockdown the expression of ATF4 led to reduced transcription of *PHGDH* and decreased intracellular serine (Fig. 1g, h). These results indicate that PHGDH-mediated serine biosynthesis is downregulated during cellular senescence.

### PHGDH-mediated serine biosynthesis prevents premature cellular senescence

To investigate the effect of serine on endothelial cell senescence, we cultured HUVECs in serine-depleted ECM medium (serine-free medium) and then treated cells with serine or glycine. Exogenous serine but not glycine dose dependently reduced the percentage of senescent cells, which was shown by reduced senescence-associated β-galactosidase (SA-β-gal) staining[24] and decreased senescence-associated heterochromatin foci (SAHF) compaction as indicated by 4′, 6-diamidino-2-phenylindole (DAPI) staining[25] (Fig. 1i; Supplementary Fig. 1g). The expression of senescence-induced marker, cyclin-dependent kinase inhibitor p21 was also reduced by serine in a dose-dependent manner (Supplementary Fig. 1h, i). Serine not only attenuated the senescence phenotype in young (P10) HUVECs but also in replicative senescent (P20) HUVECs (Fig. 1i). In addition, serine but not glycine reduced the senescence phenotype in quiescent HUVECs induced by contact inhibition (Supplementary Fig. 1j, k), suggesting that the effect of serine on cell senescence is independent of cell growth. Similar results were also observed in human artery ECs (HAECs) (Supplementary Fig. 1l).

Considering the specific loss of PHGDH in senescent HUVECs, we next evaluated the effect of PHGDH silencing on cellular senescence. Knockdown of PHGDH impaired the growth of HUVECs and caused a G0/G1 arrest (Supplementary Fig. 2a, b), indicative of defects in cell proliferation and cell cycle progression, consistent with the report that PHGDH is required for proliferation and survival of endothelial cells[19]. In line with these observations, PHGDH deficiency significantly increased the expression of p21 and reduced the expression of high-mobility group box protein 1 (HMGB1), whose expression has been reported to be reduced in senescent cells[26] (Fig. 1j; Supplementary Fig. 2c). Further SA-β-gal staining and SAHF assays showed the accelerated cellular senescence in PHGDH-knockdown HUVECs (Fig. 1k). Overexpression of PHGDH (PHGDH OE) delayed cellular senescence in both young (P10) and replicative senescent (P20) HUVECs (Supplementary Fig. 2d, e). Most importantly, exogenous serine amendment partly reversed the accelerated senescence phenotype caused by PHGDH silencing (Fig. 1l), indicating that PHGDH regulates senescence in part by contributing to serine biosynthesis. Knockdown of ATF4 also accelerated cellular senescence and addition of serine partly rescued the senescence phenotype (Fig. 1l). The accelerated senescence phenotype was also observed in siPSAT1- and siPSPH-transfected HUVECs (Supplementary Fig. 2f, g). Therefore, both loss- and gain-of-function studies highlight that PHGDH-mediated serine biosynthesis alleviates premature cellular senescence.

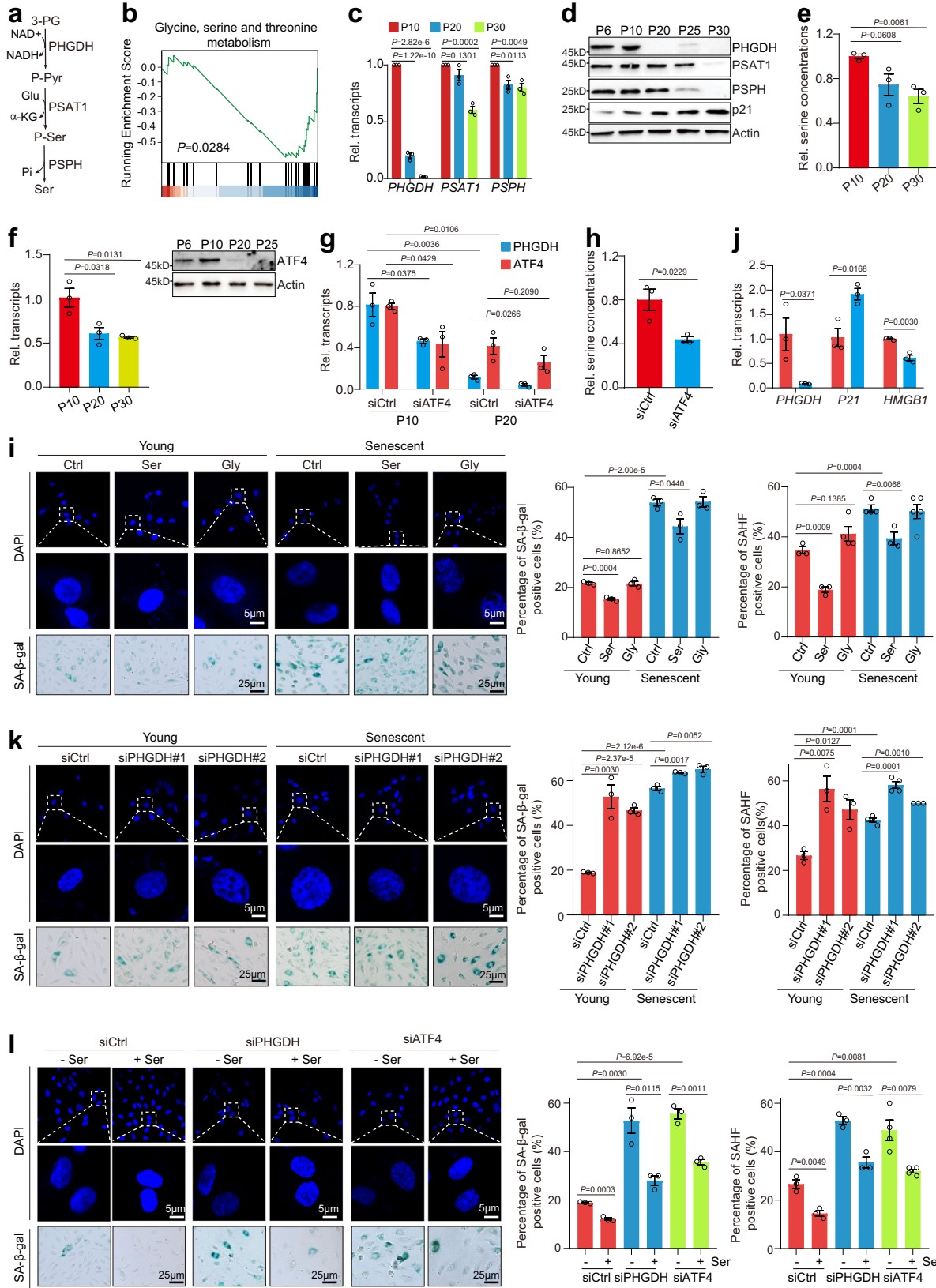

We also examined whether supplementation of the downstream products in serine metabolism pathway rescued PHGDH silencing-induced senescence. Addition of dimethyl α-ketoglutarate (DMKG), which is a membrane-permeable ester of α-ketoglutarate, a metabolite produced during serine biosynthesis pathway (Fig. 1a), did not restore the senescence phenotype of siPHGDH-transfected HUVECs

(Supplementary Fig. 2h). Exogenous supplementation of glycine failed to attenuate cell senescence in siPHGDH-transfected HUVECs (Supplementary Fig. 2i). Serine synthesis has been reported to be required for heme production in ECs[19]. However, exogenous addition of heme did not rescue the accelerated senescence in PHGDH-knockdown HUVECs (Supplementary Fig. 2j). Serine has also been converted to

**Fig. 1 | PHGDH-mediated serine synthesis pathway prevents premature HUVECs senescence. a** Diagram illustrating the de novo serine biosynthesis pathway derived from glycolysis. 3-PG, 3-phosphoglycerate; P-Pyr, 3-phosphohydroxypyruvate; P-Ser, phosphoserine; Ser, serine. **b** GSEA profiles showing senescence-regulated genes were significantly enriched in glycine, serine and threonine metabolism. Analysis of PHGDH, PSAT1 and PSPH expression in different passages of HUVECs by RT-qPCR (**c**) and immunoblots (**d**). **e** Analysis of the relative intracellular serine levels in different passages of HUVEC cells (P10, P20 and P30) when cultured in serine-free medium. **f** Analysis of ATF4 expression in HUVECs during senescence by RT-qPCR and immunoblots. **g** RT-qPCR analysis of *ATF4* and *PHGDH* transcription in siCtrl and siATF4 young (P10) and senescent (P20) HUVECs. **h** Analysis of the relative intracellular serine levels in siCtrl and siATF4 HUVECs when cultured in serine-free medium. **i** Effect of serine (Ser) and glycine (Gly) on HUVECs senescence as determined by SAHF formation (DAPI staining) and SA-β-gal staining. HUVECs (young, P10; senescent, P20) were cultured in serine-free medium

and then treated with 300 μM serine or glycine for 48 h. Right panel: Quantification of the percentage of SAHF-positive and SA-β-gal positive cells. **j** RT-qPCR analysis of *PHGDH*, *P21* and *HMGB1* transcription in siCtrl and siPHGDH HUVECs when cultured in serine-free medium. **k** Effect of PHGDH knockdown on the senescence of HUVECs (young, P10; senescent, P20) when cultured in serine-free medium as determined by SAHF formation (DAPI staining) and SA-β-gal staining. Right panel: Quantification of the number of SAHF-positive and SA-β-gal positive HUVECs. **l** Effect of serine (Ser) on HUVECs senescence in control (siCtrl), PHGDH knockdown (siPHGDH) and ATF4-knockdown (siATF4) HUVECs. Cells were cultured in serine-free medium and then treated with 300 μM serine for 24 h. For **h**, **j**, **l**, HUVECs (P10) were used in the experiments. For **c**, **e–j**, data represent means ± SE; *n* = 3 independent experiments. For **i**, **j**, *n* = 4 independent experiments. Two-sided *t*-tests were used for statistical analysis. For **d**, a typical example of three biological replicates is shown.

C16:0-ceramide in cancer cells to support cell growth[27]. Addition of C16:0-ceramide did not rescue the senescence phenotype caused by PHGDH silencing (Supplementary Fig. 2k). These data suggest that serine but not its downstream metabolites primarily prevent cellular senescence.

Vascular ECs have unique properties to form new blood vessels and spontaneous tube formation, which is important to repair ischemic myocardium or stroke[28]. We thus examined the effect of PHGDH-mediated serine biosynthesis on tube forming capability of HUVECs. Exogenous serine significantly increased tube formation in HUVECs (Supplementary Fig. 2l). Knockdown of PHGDH significantly reduced the tube formation of HUVECs (Supplementary Fig. 2m).

**Serine attenuates cellular senescence in part by targeting PKM2**

To identify the potential target of serine, we employed the drug affinity responsive target stability (DARTS) technique, which is based on the reduction in the protease susceptibility of the target proteins upon the small molecule binding[29]. We incubated serine with HeLa cell lysates, which were then digested with pronase (Fig. 2a). The proteins that bind serine will be protected from degradation by pronase. By liquid chromatography-mass spectrometry (LC-MS), we found that serine can protect pyruvate kinase M2 (PKM2) from proteolysis (Supplementary Fig. 3a), which is confirmed by immunoblots (Fig. 2b). The molecular docking experiment also showed that serine can directly bind PKM2 (Fig. 2c). These results are consistent with the reports that serine is an allosteric activator for PKM2 within the physiological concentration range[30]. As PKM2 is required to maintain endothelial cell proliferation and vascular integrity[17,18], all these data promoted us to investigate whether serine delays cellular senescence by activating PKM2.

We first examined the effect of PKM2 silencing on cellular senescence. Although the transcription of *PKM2* was marginally reduced in senescent HUVECs (Supplementary Fig. 3b), its protein level was dramatically decreased in replicative senescent HUVECs, $H_2O_2$-, and etoposide-induced senescent HUVECs (Fig. 2d; Supplementary Fig. 3c). Depletion of PKM2 by siPKM2 led to decreased cell proliferation and G0/G1 arrest (Supplementary Fig. 3d, e). Loss of PKM2 also accelerated cellular senescence as indicated by elevated expression of p21, reduced expression of HMGB1, and increased percentage of SA-β-gal positive cells and SAHF formation (Fig. 2e, f; Supplementary Fig. 3f). To examine whether the accelerated senescence in PKM2-knockdown cells was caused by increased PKM1 (the homolog of PKM2), we simultaneously depleted PKM1 and PKM2 by PKM shRNA and observed similar accelerated senescence phenotype with shPKM2 HUVECs (Supplementary Fig. 3g), indicating that PKM2 but not PKM1 regulates cellular senescence. Knockdown of PKM2 also significantly reduced the tube formation of HUVECs (Supplementary Fig. 3h). Conversely, overexpression of PKM2 attenuated senescence phenotype (Supplementary Fig. 3i). Ectopic expression of WT PKM2 but not

its kinase-dead mutant (K367M) rescued the senescence phenotype in PKM2-knockdown HUVECs (Fig. 2g; Supplementary Fig. 3j)[31], suggesting that PKM2 catalytic activity is required to alleviate cellular senescence. The histidine residue at position 464 (H464) of PKM2 is required to bind serine[30] (Fig. 2b, c). Compared with WT PKM2, ectopic expression of PKM2 H464A mutant did not ameliorate cellular senescence when grown in serine-containing medium (Fig. 2g; Supplementary Fig. 3j). Moreover, exogenous serine mitigated the senescence phenotype in control but not PKM2-knockdown HUVECs (Fig. 2h). All these data indicate that serine delays cellular senescence by targeting PKM2.

**PHGDH interacts with and protects PKM2 from autophagy-mediated degradation**

The aforementioned results showed that exogenous serine supplement is not sufficient to completely restore the accelerated senescence phenotype caused by PHGDH silencing (Fig. 1l), implying that PHGDH has a greater regulatory role in addition to contributing to serine synthesis. To gain insights into the serine-independent functions of PHGDH, we first investigated PHGDH-interacting proteins. PHGDH was immunoprecipitated and liquid chromatography-mass spectrometry (LC-MS) analysis revealed that PKM2 was a PHGDH-interacting protein (Fig. 3a). LC-MS analysis of immunoprecipitated PKM2 also showed the presence of peptides corresponding to PHGDH (Fig. 3a). To ascertain the interaction between PKM2 and PHGDH, we performed immunoprecipitation with anti-PHGDH and anti-PKM2 antibodies and observed the interaction between endogenously expressed PHGDH and PKM2 (Fig. 3b; Supplementary Fig. 4a, b). The in vitro Co-IP assay showed that the purified recombinant GST-PHGDH directly interacted with purified recombinant His-PKM2 (Fig. 3c). The immunofluorescence staining showed that PHGDH co-localized with PKM2 inside cells, especially within the nucleus (Supplementary Fig. 4c), which further confirmed the interaction between PHGDH and PKM2. Our Co-IP also showed that PKM2 and PHGDH interacted with the other two serine biosynthesis enzymes, PSPH and PSAT1 (Fig. 3b). Mutation of PKM2 H464A had no effect on the interaction between PKM2 and PHGDH (Supplementary Fig. 4d).

Interestingly, we found that knockdown of PHGDH reduced PKM2 protein level and overexpression of PHGDH increased PKM2 protein level in both HeLa and HUVECs (Fig. 3d; Supplementary Fig. 4e). The mRNA level of *PKM2* was slightly reduced relative to PKM2 protein level change in PHGDH-knockdown cells (Supplementary Fig. 4f), indicating that PHGDH primarily regulates PKM2 expression not at the transcription level. To examine whether PHGDH affects PKM2 protein stability, we blocked protein synthesis with cycloheximide (CHX) and then examined PKM2 protein level. Compared with PKM2 in control (shCtrl) cells, the half-life of PKM2 protein was significantly reduced in shPHGDH cells (Fig. 3e), indicating that PHGDH is required to maintain PKM2 stability. We also observed reduced PKM2 protein level but not

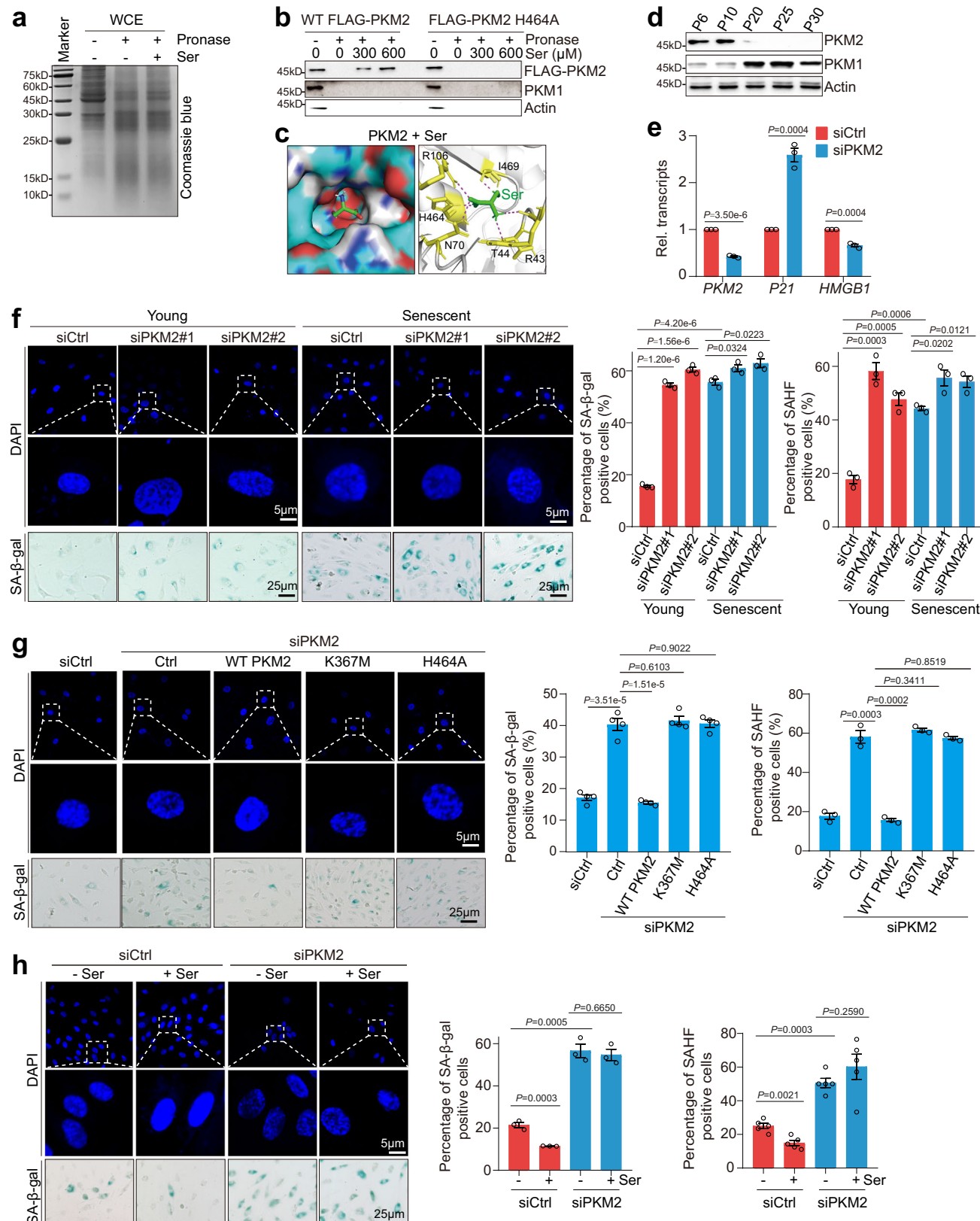

transcription level in ATF4-knockdown HUVECs (Fig. 3f; Supplementary Fig. 4f), consistent with the role of ATF4 in regulation of PHGDH expression.

There are two major quality control systems that degrade proteins in eukaryotic cells: the ubiquitin-proteasome system (UPS) and autophagy[32]. We first tested whether PKM2 is degraded by proteasome

in shPHGDH HUVECs. Treatment with the ubiquitin-proteasome inhibitor, MG132, failed to restore the reduced PKM2 protein level in shPHGDH HUVECs (Supplementary Fig. 4g). However, treatment with leupeptin, an inhibitor for autophagy-lysosome pathway, rescued the reduced PKM2 in PHGDH-knockdown HUVECs (Fig. 3g). Moreover, knockdown of the lysosomal membrane receptor LAMP2A restored

**Fig. 2 | Serine alleviates senescence of endothelial cells by targeting PKM2.**
**a** Serine bound to PKM2 as determined by DARTS. HeLa lysates were incubated with or without 300 μM serine and then subjected to pronase (50 μg/ml) digestion for 15 min. The digested products were resolved on SDS-PAGE and detected by Coomassie blue-staining. **b** PKM2 H464 is required for PKM2 to bind serine as determined by DARTS. The lysates of HUVECs (P10) that stably overexpress WT FLAG-PKM2 and FLAG-PKM2 H464A were incubated with 0–600 μM serine and then subjected to pronase (50 μg/ml) digestion for 15 min. The digested products were analyzed by immunoblots with anti-FLAG antibody. Actin was served as a loading indicator. **c** Molecular docking showing the interaction between serine (Ser, green color) and PKM2. PKM2 residues that bind serine were labeled. **d** Immunoblots of PKM2 and PKM1 in HUVECs during senescence. **e** RT-qPCR analysis of *P21* and *HMGB1* transcription in control (siCtrl) and PKM2-knockdown (siPKM2) HUVECs.

**f** Effect of PKM2 knockdown on cellular senescence in HUVECs (young, P10; senescent, P20) as determined by SAHF detection and SA-β-gal staining. **g** Effect of ectopic expression of WT PKM2 and its mutants (PKM2 K367M, H464A) on cellular senescence as determined by SAHF detection and SA-β-gal staining. HUVECs that stably overexpress PKM2 (WT PKM2, K367M, H464A) were transfected with siPKM2 to knock down the endogenous PKM2. The ectopic expressed PKM2 was resistant to siPKM2. **h** Effect of serine (Ser) on cellular senescence in control (siCtrl) and PKM2-knockdown (siPKM2) HUVECs. HUVECs were cultured in serine-free medium and then treated with 300 μM serine for 24 h. For **e, g, h**, the HUVECs at P10 were used in the experiments. For **e, f**, data represent means ± SE; $n = 3$ independent experiments. For **g**, $n = 4$ independent experiments. For **h**, $n = 5$ independent experiments. Two-sided $t$-tests were used for statistical analysis. For **b, d**, a typical example of three biological replicates is shown.

the reduced PKM2 in shPHGDH HUVECs (Supplementary Fig. 4h), indicating that PHGDH deficiency induces autophagy-mediated PKM2 degradation.

## PHGDH inhibits PCAF-catalyzed PKM2 K305 acetylation to protect PKM2 from autophagy-mediated degradation

PKM2 has been reported to be acetylated by acetyltransferase PCAF (p300/CBP-associated factor) at lysine 305 (K305), which facilitates PKM2 to be degraded by chaperone-mediated autophagy, a central lysosomal-mediated process[33]. We performed molecular docking for PKM2 and PHGDH and found that PHGDH bound to the regions (K305-R319, Y390-K422, R516-V527) of PKM2 (Fig. 3h; Supplementary Fig. 4i). As PHGDH can prevent autophagy-mediated PKM2 degradation, we hypothesized that PHGDH may inhibit the binding and acetylation of PKM2 K305 by PCAF. Indeed, knockdown of PHGDH increased PKM2 K305 acetylation (PKM2 K305ac) (Fig. 3i). We then examined the effect of PHGDH on the interaction between PCAF and PKM2. PKM2 was IPed from cells and PCAF was co-IPed with PKM2 (Fig. 3j). When PKM2 was IPed from cells overexpressing Myc-PHGDH, the amount of PHGDH co-IPed with PKM2 was increased but the amount of co-IPed PCAF was reduced (Fig. 3j). Further in vitro co-IP showed that the purified recombinant PHGDH directly reduced the interaction between PCAF and PKM2 (Fig. 3k).

We then replaced PKM2 K305 with nonacetylatable arginine (PKM2 K305R) to block K305ac and examined the stability of PKM2 K305R in PHGDH-knockdown cells. Compared with WT PKM2, PKM2 K305R mutant had a longer half-life in siPHGDH cells after incubation with cycloheximide (CHX) (Fig. 3l). Knockdown of PCAF reduced PKM2 K305ac and rescued the decreased PKM2 in shPHGDH cells (Supplementary Fig. 4j). The histone deacetylase SIRT2 is responsible for PKM2 K305 deacetylation[33,34]. Overexpression of SIRT2 also reduced PKM2 K305ac and restored the decreased PKM2 in shPHGDH cells (Supplementary Fig. 4k). These data indicate that PHGDH impairs the interaction between PCAF and PKM2 to reduce PKM2 K305ac and protect PKM2 from being degraded by autophagy.

We then examined the effect of PKM2 K305ac on cellular senescence. HUVECs that stably overexpress WT PKM2 and PKM2 K305R (to mimic deacetyl modification) were transfected with siPKM2 to knock down the endogenous PKM2. Compared with WT PKM2, PKM2 K305R mutation increased PKM2 and substantially attenuated cellular senescence (Fig. 3m). Together, these results indicate that PHGDH prevents premature senescence partly by inhibiting PCAF-catalyzed PKM2 K305ac and stabilizing PKM2.

## PHGDH enhances p300-catalyzed PKM2 K433 acetylation and promotes its nuclear translocation

PKM2 normally presents in the cytoplasm as a tetramer and acts as a metabolite kinase[31]. However, upon activation of epidermal growth factor (EGF) receptor (EGFR), PKM2 can translocate into the nucleus in a form of dimer and serve as a transcription coactivator or protein kinase in cancer cells[31]. Confocal microscopic analysis revealed that the

expression of PKM2 was decreased in shPHGDH cells (Fig. 4a), consistent with our immunoblots data. Moreover, we found the nuclear PKM2 was reduced in PHGDH-silenced HUVECs (Fig. 4a), implying that PHGDH may affect PKM2 subcellular localization. To confirm that, we performed subcellular fractionation assay for control and PHGDH-knockdown HUVECs. Immunoblots of cytoplasmic and nuclear fractions revealed the presence of PHGDH, PSAT1 and PSPH in the nucleus with PKM2 (Fig. 4b). Moreover, we observed a significant decrease of nuclear PKM2 signal in shPHGDH HUVECs (Fig. 4b), in accordance with our confocal microscopic results (Fig. 4a). Nonetheless, the nucleus localization of PSAT1 and PSPH was unaffected by knockdown of PHGDH (Fig. 4b). Overexpression of PHGDH promoted the nuclear translocation of PKM2 even in replicative senescent (P20) HUVECs (Supplementary Fig. 4l).

The acetyltransferase p300 has been reported to acetylate PKM2 at K433, which is required for PKM2 dimerization and nuclear translocation in cancer cells[35]. To further confirm the effect of PHGDH on nuclear translocation of PKM2, we analyzed PKM2 K433ac in siCtrl and siPHGDH cells using a customized anti-PKM2 K433ac antibody with high specificity (Supplementary Fig. 4m, n). Silencing of PHGDH significantly reduced PKM2 K433ac (Fig. 4c, d), consistent with reduced nuclear localization of PKM2 in shPHGDH cells. We then examined the effect of PHGDH on the interaction between p300 and PKM2. Strikingly, overexpression of PHGDH increased the interaction between PKM2 and p300, whereas knockdown of PHGDH reduced the PKM2-p300 interaction (Fig. 4e, f). Further in vitro Co-IP showed that the purified recombinant PHGDH directly increased the amount of PKM2 co-IPed with p300 (Fig. 4g), suggesting that PHGDH facilitates p300 to bind and acetylate PKM2 K433, which promotes its nuclear localization. The molecular docking also showed that PHGDH can form a stable form with PKM2 and p300 (Fig. 4h), further supporting that PHGDH facilitates p300-mediated PKM2 K433 acetylation.

As PKM2 co-localized with PHGDH within the nucleus (Supplementary Fig. 4c), we thus examined the interaction between PHGDH and PKM2 mutants (nonacetylatable PKM2 K433R, acetylation-mimetic PKM2 K433Q). PKM2 K433R and PKM2 K433Q have been reported to primarily exist as tetramers and dimers, respectively[35]. Both in vivo and in vitro Co-IP showed that the interaction between PKM2 K433Q and PHGDH was higher than the interaction between PKM2 K433R and PHGDH (Fig. 4i, j). Consistently, knocking down p300 reduced the interaction between PHGDH and PKM2 (Fig. 4k), suggesting that PHGDH might prefer to interact with dimerized PKM2.

We next examined the effect of PKM2 K433R and PKM2 K433Q mutations on cellular senescence. HUVECs were infected with lentiviruses to stably overexpress WT PKM2, PKM2 K433R and PKM2 K433Q. Compared with WT PKM2 and PKM2 K433Q, PKM2 K433R mutation was localized in the cytoplasm and significantly accelerated cellular senescence (Fig. 4l; Supplementary Fig. 4o). Collectively, these data indicate that PHGDH promotes the nuclear translocation of PKM2, which prevents premature cellular senescence.

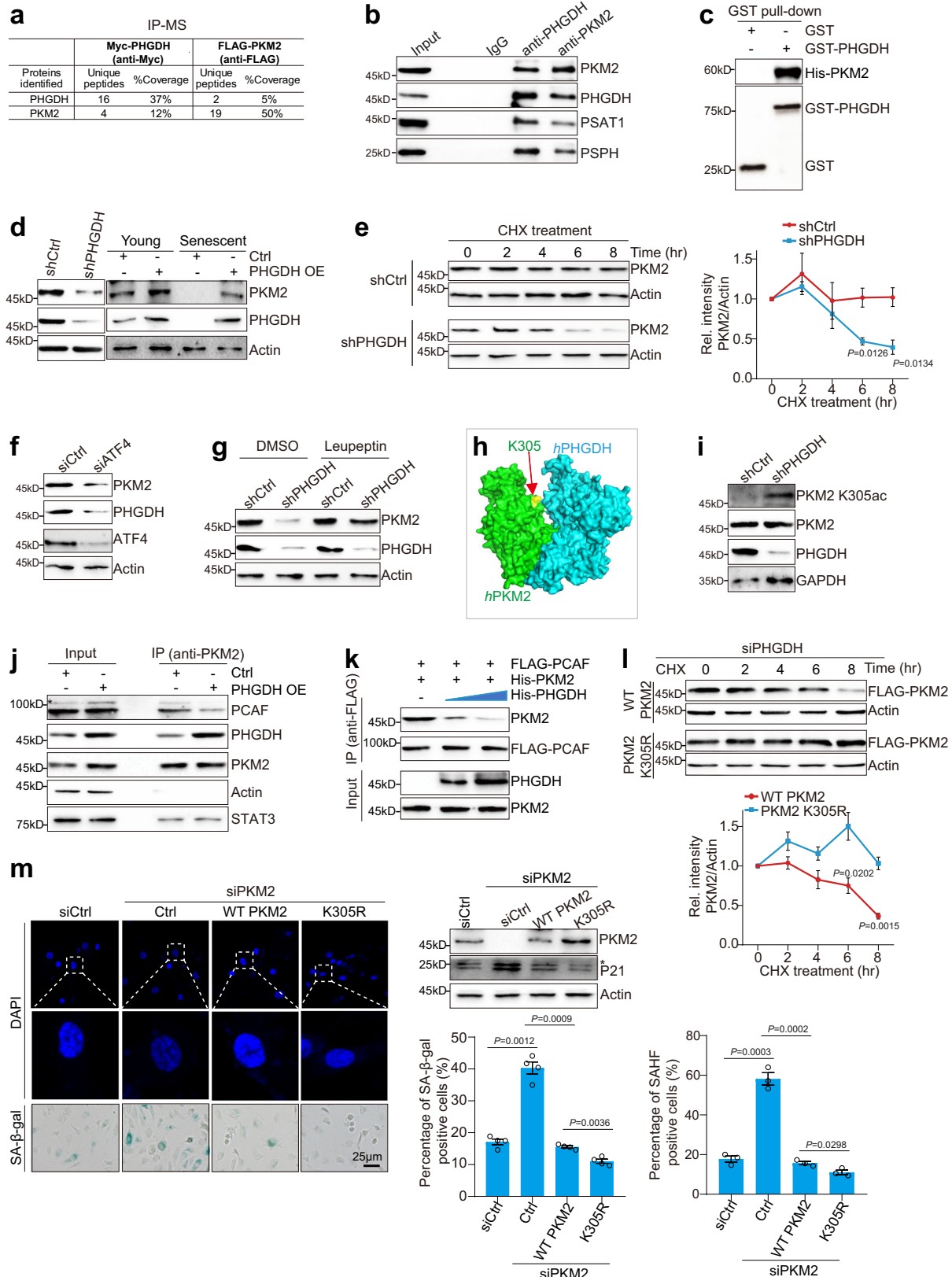

## PHGDH stimulates the activity of PKM2 to phosphorylate histone H3 at threonine 11 (H3pT11)

We next focused on the nuclear functions of PHGDH and PKM2 in cellular senescence. In tumor cells, PKM2 has been shown to translocate into the nucleus and function as a protein kinase to phosphorylate histone H3T11 (H3pT11) to regulate gene transcription and promote tumorigenesis[36]. H3pT11 was reduced in replicative senescent, $H_2O_2$-induced and etoposide-induced senescent HUVECs (Fig. 5a; Supplementary Fig. 5a). Knockdown of PKM2 significantly reduced H3pT11 in HUVECs (Fig. 5b, c; Supplementary Fig. 5b). Overexpression of WT PKM2 but not its kinase-dead mutant PKM2 K367M increased H3pT11 in HUVECs (Fig. 5d). Overexpression of WT PKM2 but not PKM2 K367M

**Fig. 3 | PHGDH prevents PCAF-mediated PKM2 K305 acetylation and autophagy-mediated PKM2 degradation. a** Mass spectrometry analysis of PHGDH- and PKM2-interaction proteins. The ectopic expressed Myc-PHGDH and FLAG-PKM2 were immunoprecipitated from HeLa cells with anti-Myc and anti-FLAG antibodies as the bait. **b** Co-IP assay showing PKM2 interacted with PHGDH, PSAT1 and PSPH. PKM2 and PHGDH were immunoprecipitated from HUVEC cells by anti-PKM2 and anti-PHGDH antibodies, respectively. **c** In vitro GST pull-down assay showing the direct interaction between purified recombinant GST-PHGDH and His-PKM2. **d** Immunoblots of PKM2 in PHGDH-knockdown (shPHGDH) and PHGDH overexpression (PHGDH OE) HUVECs. **e** PHGDH knockdown impaired PKM2 stability. Control (shCtrl) and PHGDH-knockdown (shPHGDH) HUVECs were treated with cycloheximide (CHX) (50 μg/ml) for 0–8 h. **f** Immunoblots of PHGDH and PKM2 in control (shCtrl) and ATF4-knockdown (siATF4) HUVECs. **g** Effect of autophagy inhibitor, leupeptin on PKM2 levels in control (shCtrl) and PHGDH-knockdown (shPHGDH) HUVECs. shCtrl and shPHGDH cells were treated with 20 μM leupeptin for 6 h. **h** Molecular docking showing the interaction between PKM2 and PHGDH. PKM2 K305 was indicated by an arrow. **i** Immunoblots of PKM2

K305ac in control (shCtrl) and PHGDH-knockdown (shPHGDH) HUVECs. **j** Co-IP assay showing overexpression of PHGDH (PHGDH OE) reduced the interaction between PKM2 and PCAF. Asterisk indicates non-specific bands. **k** In vitro Co-IP showing recombinant purified PHGDH decreased the direct binding of PKM2 with PCAF. FLAG-PCAF was purified from HeLa cells and incubated with purified recombinant His-PKM2 in the presence of increasing amount of purified recombinant PHGDH, which were then immunoprecipitated with anti-FLAG beads. **l** PKM2 K305R mutant was more stable than WT PKM2 in PHGDH-knockdown (siPHGDH) cells. HeLa cells that stably express WT FLAG-PKM2 and FLAG-PKM2 K305R were transfected with siPHGDH and then treated with CHX. **m** Effect of ectopic expression of WT PKM2 and PKM2 K305R mutant on cellular senescence. HUVECs that stably express FLAG-WT PKM2 and FLAG-PKM2 K305R were transfected with siPKM2 to knock down the endogenous PKM2. For **e**, **l**, data represent means ± SE; $n = 3$ independent experiments. For **m**, $n = 4$ independent experiments. Two-sided $t$-tests were used for statistical analysis. For **b**–**d**, **f**, **g**, **i**–**k**, a typical example of at least two biological replicates is shown.

---

rescued the reduced H3pT11 in PKM2-knockdown HeLa cells (Supplementary Fig. 5c). Accordingly, the lentiviral-mediated overexpression of PKM2 K305R increased PKM2 as well as H3pT11, while overexpression of PKM2 K433R reduced H3pT11 in HUVECs (Supplementary Fig. 5d), suggesting that the nuclear PKM2 phosphorylates H3T11 in endothelial cells.

As it has been established that PHGDH enhances PKM2 stability and promotes PKM2 nuclear translocation, we wondered whether PHGDH activates the activity of PKM2 to phosphorylate H3T11. Knockdown of PHGDH significantly reduced H3pT11 in HUVECs (Fig. 5e, f; Supplementary Fig. 5e). Knockdown of ATF4 significantly reduced H3pT11 in HUVECs (Fig. 5g). Conversely, overexpression of PHGDH significantly increased H3pT11 in HUVECs (Supplementary Fig. 5e)[37].

To directly show that PHGDH stimulates the protein kinase activity of PKM2 to phosphorylate H3T11, we performed the in vitro kinase assay with purified PKM2 and purified recombinant histone H3. When PKM2 was purified from HEK293T cells that express CMV promoter-driving FLAG-PKM2 and used in the in vitro kinase assay, very weak H3pT11 signal was detected (Fig. 5h, lane 2); however, when PKM2 was purified from HEK293T cells simultaneously expressing CMV promoter-driving FLAG-PKM2 and Myc-PHGDH and used in the in vitro kinase assay, strong signals for co-IPed PHGDH and H3pT11 were detected (Fig. 5h, lane 3). We also performed in vitro kinase assay using recombinant His-PKM2 and His-PHGDH expressed and purified from *E. coli*. The purified recombinant PKM2 alone had little activity on H3T11 but its activity was dramatically enhanced when pre-incubated with purified recombinant His-PHGDH (Fig. 5i; Supplementary Fig. 5f), indicating that PHGDH directly stimulates the activity of PKM2 to phosphorylate H3T11.

As serine is an activator for PKM2, we then examined the effect of serine on PKM2-catalyzed H3pT11. Treatment with serine but not glycine significantly increased the global H3pT11 in a dose-dependent manner (Supplementary Fig. 5g, h). Meanwhile, serine partly rescued the decreased H3pT11 in siPHGDH cells but not in siPKM2 cells (Fig. 5j). Serine treatment also had no effect on H3pT11 in PKM2 H464A cells (Supplementary Fig. 5i). Moreover, we performed the in vitro kinase assay with purified recombinant PKM2 in the presence or absence of serine. Adding serine remarkably enhanced the activity of PKM2 to phosphorylate H3T11 (Fig. 5k).

We then examined the effect of H3pT11 on cellular senescence. Mammalian cells contain many copies of H3 genes, it is impossible to get endogenous H3T11 mutant cells. Nonetheless, the lentiviral-mediated expression of H3T11A under CMV promoter in HUVECs reduced the global H3pT11 when compared with the lentiviral-mediated overexpression of WT H3 (Fig. 5l). Overexpression of H3T11A mutation significantly increased p21 expression, reduced

HMGB1 expression and accelerated cellular senescence when compared with overexpression of WT H3 (Fig. 5l, m; Supplementary Fig 5j). H3T11A mutation also reduced the tube formation of HUVECs (Supplementary Fig 5k).

Chk1 has been reported to phosphorylate H3T11 to regulate the expression of cell cycle genes upon DNA damage[38]. However, knockdown of Chk1 had no significant effect on cellular senescence (Supplementary Fig 5l). Collectively, all these data indicate that PHGDH ameliorates cellular senescence partly, if not completely, by promoting PKM2-catalyzed H3pT11.

### The PHGDH-PKM2-H3pT11 axis regulates the transcription of *SIRT1* and senescence-associated genes

RNA-seq analysis of PHGDH- and PKM2-knockdown cells revealed that their transcriptomes are positively correlated with a correlation efficiency of 0.42 (Fig. 6a). There are 332 co-upregulated genes and 219 co-downregulated genes in siPHGDH and siPKM2 HUVECs (Supplementary Fig. 6a, b). KEGG analysis revealed that 332 co-upregulated genes are significantly enriched in vascular smooth muscle contraction, metabolic pathway, cytokine-cytokine receptor interaction, cytosolic DNA-sensing pathway, cellular senescence, IL-17 signaling pathway, etc (Supplementary Fig. 6c). GSEA analysis revealed that PKM2 and PHGDH significantly regulate genes associated with cellular senescence (Fig. 6b). Based on the published databases, including the CSGene and Ingenuity Pathway Analysis databases[39,40], we have identified 639 genes implicated in cellular senescence[14]. Among these senescence-associated genes, 89 genes and 30 genes were differentially regulated by PHGDH and PKM2 in HUVECs, respectively (Fig. 6c). PHGDH and PKM2 co-regulate the expression of 21 senescence-related genes (Fig. 6d). RT-qPCR confirmed that knockdown of PHGDH and PKM2 significantly reduced the transcription of senescence-associated genes, including *SIRT1*, *HIST1H3H*, *BMI1*, and *LAMA1* (Fig. 6e). The transcription of these genes was also significantly reduced in siATF4 HUVECs (Fig. 6f).

Among these differentially regulated genes, SIRT1 has been reported to play essential roles in regulating ageing and related diseases[41]. The expression of SIRT1 was significantly decreased in replicative senescent, $H_2O_2$-induced and etoposide-induced senescent HUVECs (Supplementary Fig. 7a–c). Knockdown of PHGDH significantly reduced both mRNA and protein levels of SIRT1, while overexpression of PHGDH increased SIRT1 expression (Supplementary Fig. 7d–f). Likewise, knockdown of PKM2 significantly reduced the expression of SIRT1 and overexpression of WT PKM2 but not PKM2 K367M increased SIRT1 expression (Supplementary Fig. 7g–i). Recent study has demonstrated that SIRT1 is degraded by autophagy during senescence[42]. However, blocking autophagy by leupeptin treatment had no effect on the reduced SIRT1 protein levels in PHGDH-

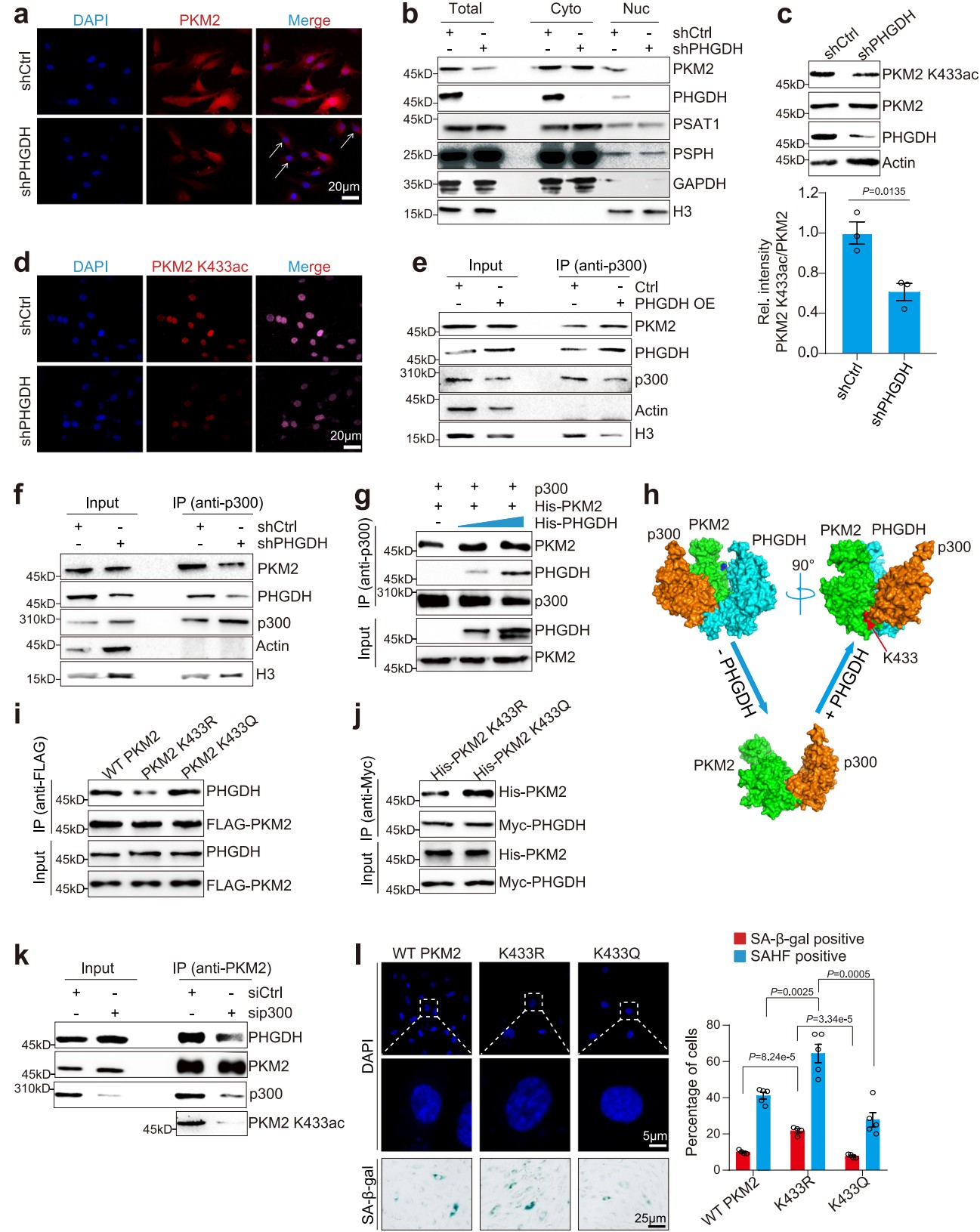

knockdown HUVECs (Supplementary Fig. 7j) and PKM2-knockdown HUVECs (Supplementary Fig. 7k), indicating that the expression of SIRT1 is regulated by PHGDH and PKM2 at the transcription level in endothelial cells.

Exogenous serine induced the expression of senescence-associated genes, *SIRT1*, *BMI1* and *LAMA1* in a dose-dependent manner (Fig. 6g; Supplementary Fig. 7l, m). Serine treatment partly rescued the reduced transcription of *SIRT1*, *BMI1* and *LAMA1* in PHGDH-knockdown but not PKM2-knockdown cells (Fig. 6g). Mutation of PKM2 H464A also abolished serine-induced transcription of *SIRT1*, *BMI1* and *LAMA1* (Supplementary Fig. 7n), suggesting that PHGDH-mediated serine biosynthesis facilitates the expression of senescence-

**Fig. 4 | PHGDH enhances p300-catalyzed PKM2 K433 acetylation and promotes its nuclear location. a** Immunofluorescence analysis of the effect of PHGDH knockdown on PKM2 subcellular localization in HUVECs. Blue: DAPI; Red: PKM2. **b** Effect of PHGDH knockdown on the amount of PKM2, PSAT1 and PSPH localized in the cytoplasm (Cyto) and nucleus (Nuc) of HUVECs as determined by subcellular fractionation analysis. Effect of PHGDH knockdown (shPHGDH) on PKM2 K433ac in HUVECs as determined by immunoblots (**c**) and immunofluorescence (**d**). More lysates from PHGDH-knockdown HUVECs were loaded to keep PKM2 at similar levels. Effect of PHGDH overexpression (**e**) and PHGDH knockdown (**f**) on the interaction between PKM2 and p300 in HUVECs as determined by Co-IP. To ensure equal loading of PKM2, less cell lysate from PHGDH-overexpression (PHGDH OE) HUVECs (**e**) and more lysate from PHGDH-knockdown (shPHGDH) HUVECs (**f**) were used for immunoprecipitation. Actin and histone H3 were used as loading controls. **g** Purified recombinant PHGDH increased the direct binding of PKM2 with p300 as determined by in vitro Co-IP. p300 was immobilized on agarose beads from HeLa cells with anti-p300 and then incubated with purified recombinant PKM2 in the presence of increasing amount of purified recombinant PHGDH. **h** Molecular docking showing the formation of PHGDH/PKM2/p300. **i** PKM2 K433R decreased its interaction with PHGDH as determined by Co-IP. HUVECs were individually infected with lentiviruses to stably express WT FLAG-PKM2, FLAG-PKM2 K433R and FLAG-PKM2 K433Q. FLAG-PKM2 was IPed with anti-FLAG antibody. **j** In vitro Co-IP showing the recombinant purified His-PKM2 K433R mutant had reduced interaction with PHGDH (Myc-PHGDH) when compared with His-PKM2 K433Q. **k** p300 knockdown (sip300) decreased the binding of PKM2 with PHGDH in HUVECs. **l** Effect of ectopic expression of WT PKM2, PKM2 K433R and PKM2 K433Q on cellular senescence. HUVECs were infected with lentiviruses to stably express WT FLAG-PKM2, FLAG-PKM2 K433R and FLAG-PKM2 K433Q. For **a**–**f**, **k**, **l**, HUVECs (P10) were used in the experiments. For **c**, data represent means ± SE; $n = 3$ independent experiments. For **l**, data represent means ± SE; $n = 5$ independent experiments. Two-sided $t$-tests were used for statistical analysis. For **a**, **b**, **d**–**g**, **i**–**k**, a typical example of three biological replicates is shown.

associated genes by activating PKM2. The expression of SIRT1, BMI1 and LAMA1 was also significantly reduced in H3T11A mutant (Fig. 6h, i). In addition, ChIP analysis showed that knockdown of PHGDH or PKM2 significantly reduced the enrichment of H3pT11 at *SIRT1*, *BMI1* and *LAMA1* (Fig. 6j). Exogenous serine increased the enrichment of H3pT11 at *SIRT1*, *BMI1* and *LAMA1* in HUVECs that overexpress WT PKM2 but not PKM2 H464A mutant (Supplementary Fig. 7o), suggesting that the PHGDH-PKM2-H3pT11 axis regulates the transcription of senescence-associated genes in ECs.

We then performed RNA-seq for control (siCtrl) and siSIRT1 HUVECs. By comparing the RNA-seq data for siPHGDH, siPKM2 and siSIRT1, we found that the transcriptome of siSIRT1 positively correlated with the transcriptomes of siPHGDH and siPKM2 with a correlation efficiency of 0.58 and 0.38, respectively (Fig. 6k). There are 412 and 126 genes downregulated by PHGDH/SIRT1 and PKM2/SIRT1, respectively (Fig. 6l; Supplementary Fig. 6d). KEGG analysis of these downregulated genes revealed significant enrichment of genes associated with cellular senescence (Fig. 6m). We then overexpressed SIRT1 in PHGDH-knockdown and PKM2-knockdown cells and found that overexpression of SIRT1 (SIRT1 OE) reduced the accelerated cell senescence in siPHGDH and siPKM2 HUVECs (Fig. 6n). Altogether, these data indicate that PHGDH stimulates the ability of PKM2 to phosphorylate H3T11 and regulate the transcription of senescence-associated genes, which prevents premature cellular senescence.

### PHGDH and PKM2 ameliorate ageing features in mice

To determine whether PHGDH and PKM2 are involved in ageing process, we analyzed the expression of proteins in the PHGDH-PKM2-H3pT11 axis in mice following natural ageing. We dissected the heart tissues from young and aged C57BL/6 mice. The protein levels of ATF4, PHGDH and PKM2 were significantly reduced in the hearts of aged (12, 20, 26 months old) mice (Fig. 7a, b; Supplementary Fig. 8a). Consistent with reduced PHGDH and PKM2, the levels of H3pT11 and SIRT1 were also significantly reduced in the hearts of aged mice (Fig. 7a, b; Supplementary Fig. 8a). The relative serine concentrations were also significantly reduced in the hearts of aged mice (Fig. 7c). In addition, these proteins and H3pT11 were decreased in the liver and kidney tissues of aged mice (Supplementary Fig. 8b, c). PKM2 K305ac was significantly increased in the hearts of aged mice, whereas PKM2 K433ac was significantly reduced in the hearts of aged mice (Fig. 7d), consistent with the trend of PKM2 changes during ageing. By analyzing mRNA from the hearts of 6-month-old mice and 20-month-old mice, the transcription of *ATF4*, *PHGDH* and *SIRT1* was significantly reduced while the transcription of *PKM2* was not significantly affected during ageing (Fig. 7e), consistent with reduced transcription of *ATF4*, *PHGDH* and *SIRT1* as well as increased PKM2 degradation in senescent HUVECs. These results indicate that the PHGDH-PKM2-H3pT11 axis was downregulated in naturally aged mice.

To assess whether overexpression of PHGDH and PKM2 could delay aging in vivo, we generated a recombinant adenovirus-associated virus serotype 1 (rAAV1) cassette with *PHGDH* gene expression driven by a synthetic *ICAM2* promoter (AAV-PHGDH), which ensures vascular endothelium (VE)-specific expression[43]. We also generated a recombinant rAAV1 cassette with *PKM2* gene expression driven by a *ICAM2* promoter (AAV-PKM2). Both AAV-PHGDH and AAV-PKM2 particles were injected via tail vein into old (18 months old) mice, when the expression of PHGDH and PKM2 was dramatically reduced (Fig. 7f; Supplementary Fig. 8a). 4 weeks later, the ectopic expression of PHGDH and PKM2 in the aorta was detected by immunoblots (Fig. 7g). The expression of Myc-PHGDH and FLAG-PKM2 was detected in the endothelium and aortae in AAV-PHGDH and AAV-PKM2 mice, respectively (Fig. 7g; Supplementary Fig. 8d). After the endothelium was rubbed from the aortae of AAV-PHGDH and AAV-PKM2 mice, FLAG-PHGDH and FLAG-PKM2 cannot be detected (Fig. 7g; Supplementary Fig. 8d), suggesting the endothelium-specific expression of PHGDH and PKM2. Moreover, the protein levels of PKM2 were increased in the aorta endothelium of AAV-PHGDH mice but not in the endothelium-rubbed aortae (Supplementary Fig. 8d). We also examined the expression of AAV-PHGDH and AAV-PKM2 in the brain and trace amount was detected (Supplementary Fig. 8e), suggesting that the brain endothelial cells are mildly affected. With the VE-specific expression of PHGDH and PKM2, H3pT11 and SIRT1 were also significantly increased as indicated by immunoblots and histological staining of cardiac tissue (Fig. 7g; Supplementary Fig. 8f). Moreover, VE-specific expression of PHGDH and PKM2 significantly reduced p21 protein levels (Fig. 7g; Supplementary Fig. 8f), indicating that the PHGDH-PKM2-H3pT11 axis was enhanced in the aortae of AAV-PHGDH- and AAV-PKM2-treated mice.

Endothelial dysfunction is a causal factor of systemic aging, including dilated cardiomyopathy, reduced running endurance and cardiac functions (heart rate, cardiac output, left ventricular ejection fraction, and fractional shortening)[43]. We thus examined whether VE-specific expression of PHGDH (AAV-PHGDH) and PKM2 (AAV-PKM2) could ameliorate the ageing phenotype. Although VE-specific expression of PHGDH and PKM2 had no significant effect on body weight (Fig. 7h), the heart weight and the heart weight/body weight ratio were significantly decreased by VE-targeted expression of PHGDH and PKM2 (Fig. 7i; Supplementary Fig. 9a), indicating that PHGDH and PKM2 prevent dilated cardiomyopathy[44]. To assess the effect of VE-targeted expression of PHGDH and PKM2 on endothelial functions, we first isolated the aortae from AAV-PHGDH and AAV-PKM2 mice and examined their response to acetylcholine. VE-targeted expression of PHGDH and PKM2 significantly increased endothelium-dependent relaxation when compared with AAV-Ctrl (Fig. 7j). VE-specific expression of PHGDH and PKM2 increased collagen content in the heart (Supplementary Fig. 9b), which is consistent with the reports that

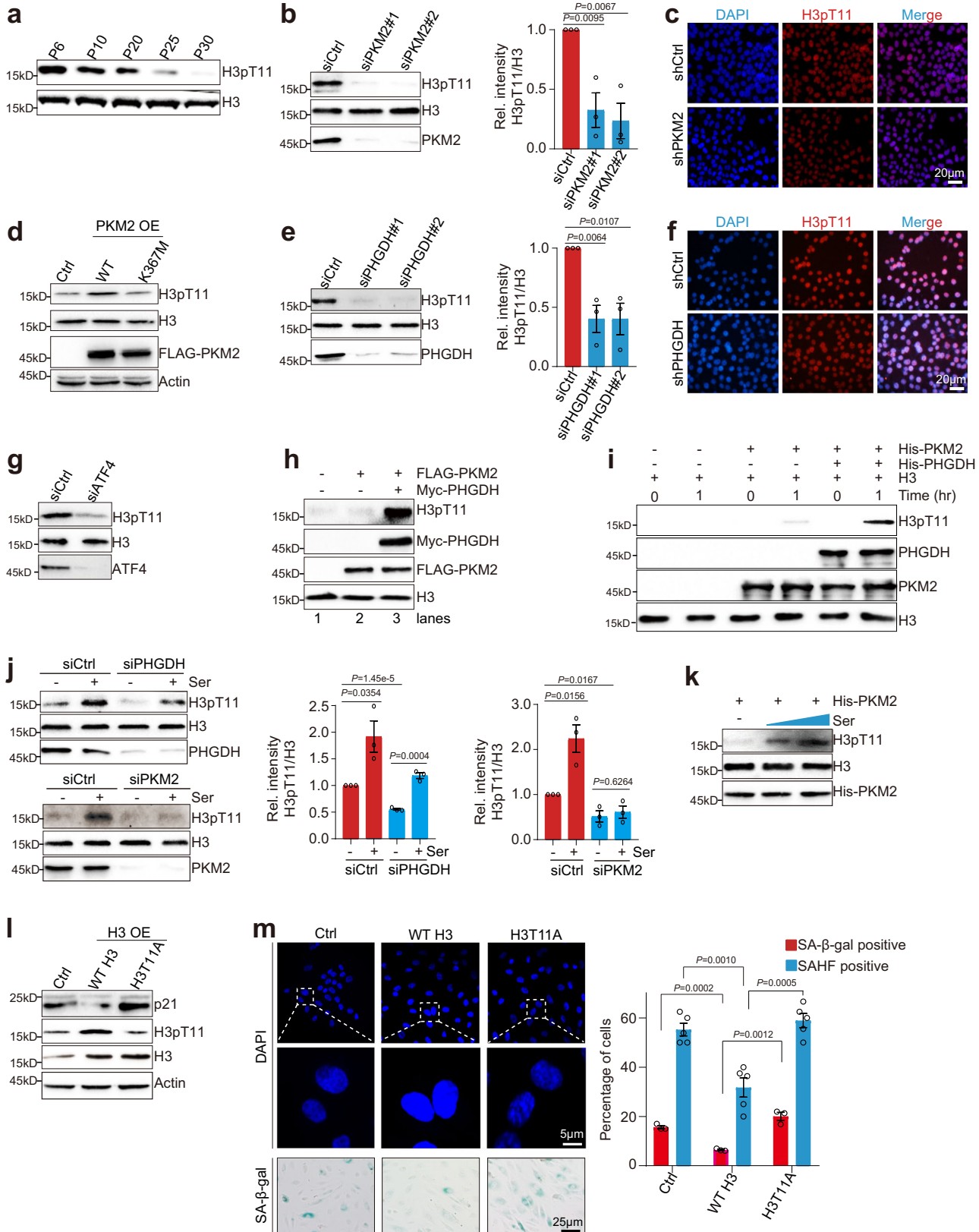

PKM2 dimer promotes glycine biosynthesis to facilitate collagen accumulation in myofibroblasts[45]. We then measured the effect of VE-targeted expression of PHGDH and PKM2 on arterial blood pressure. AAV-PHGDH and AAV-PKM2 mice had similar systolic pressure and diastolic pressure as AAV-Ctrl (Supplementary Fig. 9c, d). Echocardiographic imaging analysis showed that although AAV-PHGDH and

AAV-PKM2 mice had no significantly changed heart rate (Supplementary Fig. 9e), they had increased left ventricular (LV) ejection fraction and LV fractional shortening (Fig. 7k, l).

AAV-PHGDH and AAV-PKM2 mice were able to run 2- to 3-fold as much as their wild-type littermates (AAV-Ctrl) when subjected to a treadmill running capacity test (Fig. 7m; Supplementary Fig. 9f).

**Fig. 5 | PHGDH enhances PKM2-catalyzed H3pT11 and prevents premature cellular senescence. a** Immunoblots of H3pT11 in HUVECs during cellular senescence. Analysis of H3pT11 in control (shCtrl) and PKM2-knockdown (shPKM2) HUVECs as determined by immunoblots (**b**) and immunofluorescence (**c**). **d** Lentiviral-mediated overexpression of WT PKM2 but not PKM2 K367M mutant increased H3pT11 in HUVECs. Analysis of H3pT11 in control (shCtrl) and PHGDH-knockdown (shPHGDH) HUVECs when cultured in serine-free medium as determined by immunoblots (**e**) and immunofluorescence (**f**). **g** Immunoblots of H3pT11 in control (siCtrl) and ATF4-knockdown (siATF4) HUVECs. **h** In vitro kinase assay showing PHGDH enhanced the activity of PKM2 to phosphorylate H3T11. FLAG-PKM2 was purified from control or PHGDH-overexpression (Myc-PHGDH) HeLa cells with anti-FLAG beads. The in vitro kinase assay was performed with purified FLAG-PKM2 and recombinant purified histone H3. H3pT11 was detected by immunoblots. **i** In vitro kinase assay showing recombinant His-PHGDH directly stimulated the activity of recombinant His-PKM2 to phosphorylate H3T11. **j** Effect of serine (Ser) on H3pT11 in control (siCtrl), PHGDH-knockdown (siPHGDH) and PKM2-knockdown (siPKM2) HUVECs. **k** In vitro kinase assay showing serine (Ser) directly stimulates the activity of recombinant His-PKM2 to phosphorylate H3T11. Overexpression of WT H3 but not H3T11A mutant alleviated cellular senescence of HUVECs as determined by expression of p21 (**l**), SAHF detection and SA-β-gal staining (**m**) in HUVECs. HUVECs were infected with lentiviruses to overexpress WT H3 and H3T11A. For **b**, **e**, **j**, data represent means ± SE; $n = 3$ independent experiments. For **m**, $n = 5$ independent experiments. Two-sided $t$-tests were used for statistical analysis. For **a**, **c**, **d**, **f-i**, **k**, a typical example of two biological replicates is shown.

VE-targeted expression of PHGDH and PKM2 up-regulated the expression of key skeletal muscle proteins involved in metabolism, including peroxisome proliferator-activated receptor-γ (PPARγ), and its coactivator PGC-1α (Supplementary Fig. 9g, h). The levels of plasma adiponectin were also significantly increased in AAV-PHGDH and AAV-PKM2 mice (Supplementary Fig. 9i). Moreover, the enzymes involved in ROS control, including SOD1 and SOD2 were also up-regulated in AAV-PHGDH and AAV-PKM2 mice (Supplementary Fig. 9g, h).

To examine the above effects were caused by reduction of senescent endothelial cells or senolytic effects, we first performed SA-β-gal staining of the aortae from AAV-Ctrl, AAV-PHGDH and AAV-PKM2 mice. Overexpression of PHGDH and PKM2 significantly decreased SA-β-gal staining signal (Supplementary Fig. 9i). We then examined the senolytic effect from over-expression of PKM2 and PHGDH. The expression of pro-apoptotic protein, pro-caspase 3 and cleaved caspase 3 was unchanged among the aortae from AAV-Ctrl, AAV-PHGDH and AAV-PKM2 mice (Fig. 7g; Supplementary Fig. 8d). All these data suggest that the transfection slowed cell senescence with little senolytic effect.

Collectively, these data suggest that VE-targeted expression of PHGDH and PKM2 prevents the age-associated declines in cardiac and muscle function in vivo.

## Discussion

Although vascular endothelial cells (ECs) locate in proximity to circulating oxygen and nutrients, they oxidize only a minor fraction of glucose in the mitochondria and rely on glycolysis to generate ATP, which is quite different from other health cell types[15,16]. However, for most glycolytic enzymes and intermediates, their precise functions in endothelial biology remain to be determined. Here, we identify the downregulation of PHGDH-mediated serine biosynthesis as a hallmark of senescent ECs and find that PHGDH-mediated serine biosynthesis plays a critical role in preventing premature ECs senescence. Moreover, we uncover two mechanisms by which PHGDH delays cellular senescence (Fig. 8). On one hand, PHGDH contributes to serine biosynthesis, which then acts as a coactivator to stimulate the activity of glycolytic enzyme PKM2. On the other hand, PHGDH promotes the stability and nuclear localization of PKM2, which then phosphorylates H3T11 and regulates the expression of senescence-associated genes. Hence, our study shed lights on the function of glycolysis and glycolysis-derived serine metabolism in endothelial cellular senescence.

Exogenous serine and PHGDH-mediated serine biosynthesis play important roles in promoting tumor cell growth, regulating adaptive immunity and antiviral innate immunity[46–48]. PHGDH-mediated serine synthesis also facilitates the proliferation of ECs by contributing to heme synthesis[19]. However, it remains enigmatic about why ECs require enzymatically active PHGDH, even in the presence of external serine[19]. Here, we identify a role of PHGDH-mediated serine biosynthesis in regulating cellular senescence in addition to proliferation.

We find that PHGDH is specifically lost and the intracellular serine is gradually reduced during EC senescence. The reduced *PHGDH* transcription during senescence is regulated by transcription factor ATF4, which is in accordance with the fact that PHGDH is regulated at the transcriptional level depending on tissue specificity and cellular proliferative status[49]. Depletion of PHGDH leads to reduced cell growth, caused cell cycle arrest and accelerated senescence-associated phenotype. Overexpression of PHGDH can increase the protein level of PKM2, promote its nuclear translocation and attenuate cellular senescence in both young and senescent HUVECs. In particular, we find that serine but not its downstream metabolites, i.e. glycine, heme, C16:0-ceramide supplements attenuate ECs senescence, indicating that serine per se is able to alleviate cellular senescence, which is different with the reports that it is the downstream serine metabolites, i.e., SAM that mediate antiviral innate immunity[47]. This function is also different with the report by Vandekeere and colleagues that PHGDH-mediated serine synthesis plays an essential role in endothelial cell growth by contributing to heme synthesis, which was performed with very young HUVECs (passages 1–4, P1-P4)[19]. We used HUVECs (passages 10–20, P10-P20) to study the role of PHGDH-mediated serine biosynthesis in regulating cellular senescence. In addition, PHGDH-mediated serine biosynthesis pathway prevents premature senescence in quiescent HUVEC and human artery ECs (HAECs). The distinct effects of serine and glycine have been reported in other cellular processes: serine stimulated tumor cell proliferation, whereas glycine inhibited proliferation[50]. We do not think the senescence observed in serine-free medium is caused by a global repression of protein synthesis as decreasing the protein biosynthesis rate has been shown to extend life span[51,52]. We also identify PKM2 as the target for serine to regulate ECs senescence. Although we cannot exclude other possible targets, serine regulates cellular senescence primarily by activating PKM2. Moreover, vascular endothelium (VE)-specific expression of PHGDH and PKM2 improved the cardiac functions of aged mice, suggesting that PHGDH-mediated serine biosynthesis pathway may become potential targets to ameliorate ageing.

It has been reported that D-serine but not L-serine alleviates the age-related deficit in synaptic plasticity[53]. However, D-serine was predominantly found in the brain tissues and its amount was very low in other organs, such as liver, kidney, serum and spinal cord[54]. In contrast, L-serine ubiquitously exists in these organs[54]. It is possible that L-serine but not D-serine has a more important role in regulating the aging of these organs. Moreover, L-serine but not D-serine supplementation has been reported to cause lifespan extension in *C. elegans*[55]. This lifespan extension appears to be caused by altered mitochondrial metabolism and activation of stress response pathways, including DAF-16/FOXO and SKN-1/Nrf2[55]. It remains unknown whether it is L-serine or its metabolism mediates this effect. Here, we report that L-serine per se but not its metabolism attenuates senescence in endothelial cells by acting as a signal molecule to activate PKM2. Moreover, we identify a function of serine biosynthetic enzyme PHGDH in regulating cellular senescence.

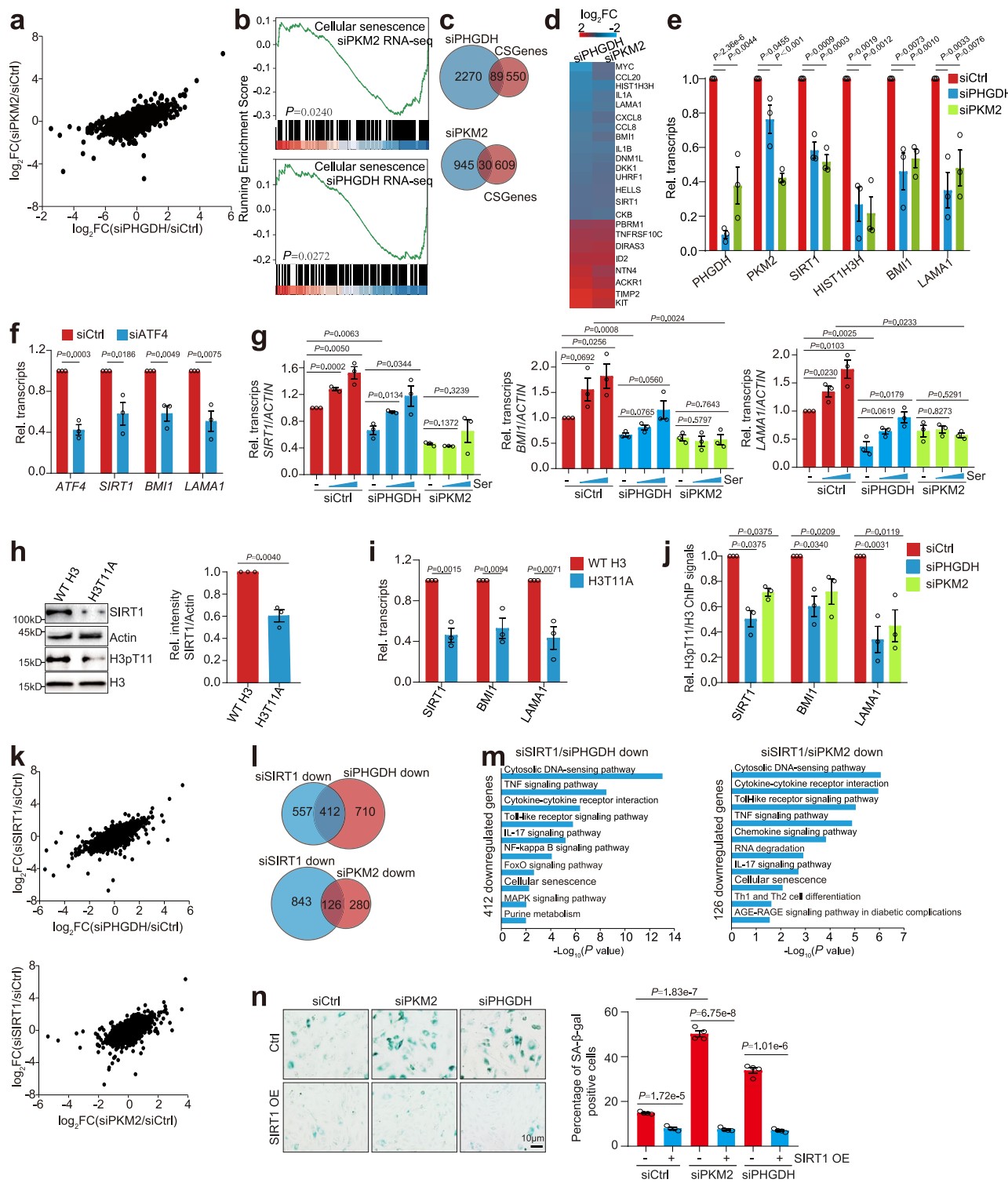

We noticed that extracellular serine cannot entirely compensate for the biological consequences of PHGDH knockdown as PHGDH is required to prevent premature senescence even in the presence of environmental serine. This partial rescue of PHGDH deficiency by exogenous serine has also been observed in regulating IFN-β production during antiviral innate immunity[47]. Supplementation of serine cannot rescue the proliferation defects in PHGDH-knockdown cells[19]. All these data indicate that PHGDH has other functions independent on serine biosynthesis. In this study, we identify an interaction between PHGDH and PKM2 and underscore the biological significance for this interaction. First, by interacting with PKM2, PHGDH and two

other serine biosynthesis enzymes, PSAT1 and PSPH may provide a local high concentration of serine for PKM2 in the nucleus to stimulate its protein kinase activity (Fig. 8), which conforms the "local production and local action model"[56]. In line with our observation, Vandekeere et al. also proposed that the reason why ECs require enzymatically active PHGDH even in the presence of external serine may be related to compartmentalization of de novo synthesized serine[19]. By binding to the region (K305-R319) of PKM2, PHGDH causes steric hindrance for PCAF to acetylate PKM2 K305, which prevents autophagy-mediated PKM2 degradation. Although PHGDH inhibits PCAF-catalyzed PKM2 K305 acetylation, PHGDH promotes p300 to acetylate PKM2 at K433.

**Fig. 6 | PKM2-catalyzed H3pT11 regulates the transcription of senescence-associated genes. a** A scatter plot from RNA-seq data showing a positive correlation between transcription changes upon PHGDH knockdown (siPHGDH, x-axis), and transcription changes upon PKM2 knockdown (siPKM2, y-axis) in HUVECs. **b** GSEA profiles showing PKM2- and PHGDH-regulated genes were enriched in cellular senescence. **c** Top panel: Venn diagrams showing the overlap between genes regulated by PHGDH and 639 senescence-associated genes. Bottom panel: Venn diagrams showing the overlap between genes regulated by PKM2 and 639 senescence-associated genes. **d** Heatmap showing the log$_2$ fold change (FC) of 21 senescence-associated genes co-regulated by PHGDH and PKM2. **e, f** RT-qPCR analysis of 4 senescence-associated genes (*SIRT1, HIST1H3H, BMI1, LAMA1*) in control (siCtrl), PHGDH- knockdown (siPHGDH), PKM2-knockdown (siPKM2) and ATF4-knockdown (siATF4) HUVECs. **g** RT-qPCR analysis the effect of serine (Ser) on transcription of *SIRT1, BMI1* and *LAMA1* in siCtrl, siPHGDH and siPKM2 HUVECs. **h** Immunoblots of SIRT1 in HUVECs that overexpress WT H3 and H3T11A. **i** RT-qPCR

analysis of *SIRT1, BMI1* and *LAMA1* transcription in HUVECs that overexpress WT H3 and H3T11A. **j** ChIP-qPCR analysis of H3pT11 occupancy at *SIRT1, BMI1* and *LAMA1* promoters in siCtrl, siPHGDH and siPKM2 HUVECs. **k** Top panel: A scatter plot from RNA-seq data showing a positive correlation between transcription changes in siPHGDH HUVECs (x-axis), and transcription changes in siSIRT1 HUVECs (y-axis). Bottle panel: A scatter plot from RNA-seq data showing a positive correlation between transcription changes in siPKM2 HUVECs (x-axis), and transcription changes in siSIRT1 HUVECs (y-axis). **l** Venn diagrams showing the overlap between genes regulated by SIRT1/PKM2, SIRT1/PHGDH. **m** KEGG analysis of 412 genes downregulated by SIRT1 and PHGDH, 126 genes downregulated by SIRT1 and PKM2. **n** Overexpression of SIRT1 (SIRT1 OE) rescued the accelerated senescence phenotype in siPKM2 and siPHGDH HUVECs. HUVECs that stably overexpress SIRT1 were transfected with siCtrl, siPKM2 and siPHGDH. For **e–j**, data represent means ± SE; *n* = 3 independent experiments. For **n**, *n* = 4 independent experiments. Two-sided *t*-tests were used for statistical analysis.

Using structure simulation and Co-IP assays, we showed that PHGDH directly facilitates p300 to interact with and acetylate PKM2 at K433, which promotes PKM2 dimerization and nuclear translocation. Thus, we identify PHGDH as a regulator of PKM2 stability and activity, which also coordinates glycolysis with de novo serine biosynthesis during senescence and ageing. Given the important role of PHGDH and PKM2 in cancer cell proliferation, PHGDH could also promote tumorigenesis by interacting and regulating the stability and activity of PKM2.

As a rate-limiting enzyme during glycolysis, there are four mammalian pyruvate kinase (PK) isoenzymes (M1, M2, L, and R), which are present in different cell types[57,58]. PKM1 is a constitutively active form of PK that is found in normal adult cells[57,58]. In contrast, PKM2 is found predominantly in the fetus and tumor cells, where the abundance of other isoforms of PK is low[57,58]. PKM2 can exist in either tetramers or dimers, but it predominantly occurs in dimers in tumor cells upon growth factor stimulation[31]. Little is known about the role of PKM2 in ageing and the underlying mechanism. Here, we find that PKM2 is degraded by autophagy during senescence, which is in contrast to regulation of *PHGDH* transcription. Reduced expression of PKM2 accelerates cellular senescence. Although PKM1 is increased during senescence, PKM2 deficiency induces senescence independent on PKM1 as loss of PKM1 and PKM2 still accelerated cellular senescence. PHGDH enhances PKM2 stability and promotes its nucleus translocation. By performing the in vitro kinase assay, we showed that PKM2 phosphorylates H3T11 in endothelial cells and provided the evidence that serine and PHGDH directly stimulates the activity of PKM2 to phosphorylate H3T11 in mammals. By overexpressing H3T11A mutant in HUVECs, we identified H3pT11 as a histone marker that delays cellular senescence. Compared with cancer cells, the lower histone levels in HUVECs enables us to study H3pT11 by overexpressing H3T11A mutant. Although PKM2 has been reported to maintain intracellular serine levels[22], we do not think the accelerated cellular senescence in PKM2-knockdown HUVECs is due to reduced serine as exogenous serine has no effect on the senescence of siPKM2 HUVECs (Fig. 2h). By phosphorylating H3T11, the nuclear PKM2 acts as a transcriptional cofactor to regulate the transcription of senescence-associated genes, including *SIRT1* (Fig. 6e). Loss of PHGDH and PKM2 has been reported to cause mitochondrial dysfunction and increased reactive oxygen species[17,19]. SIRT1 has been reported to antagonize oxidative stress and inhibit endothelial senescence by repressing the expression of protein p66Shc and inducing the expression of anti-oxidative enzymes[59,60]. It is thus possible that the PHGDH-PKM2-H3pT11 axis could also regulate mitochondrial function and oxidative stress via SIRT1. Moreover, this axis also exists during ageing as the levels of PHGDH, PKM2 and H3pT11 are reduced in aged and progeria mice. Vascular endothelium-specific overexpression of PHGDH and PKM2 increases H3pT11 and SIRT1 and attenuate the ageing phenotype in mice. Thus, our data provide a potential gene therapy to improve ageing and associated symptoms.

One important target for metabolic intervention is SIRT1, which is an NAD$^+$-dependent deacetylase that plays important roles in delaying premature senescence. SIRT1 has been reported to be degraded by autophagy in fibroblast cells during senescence[42]. Here, we find that SIRT1 could also be regulated at the transcriptional level in endothelial cells during senescence. The reduced PHGDH results in less stable and less active PKM2, leading to decreased H3pT11 and reduced SIRT1 expression. Moreover, PKM2 and PHGDH promote the expression of SIRT1 independent of autophagy. Our results indicate that PHGDH-PKM2-H3pT11 primarily regulates the transcription of *SIRT1*, which reveals an epigenetic mechanism to regulate SIRT1 expression. Meanwhile, the regulation of SIRT1 by PHGDH-mediated serine biosynthesis connects serine metabolism with SIRT1, which provides direct evidence that serine metabolism can promote SIRT1 expression.

We used a recombinant adenovirus-associated virus to express PHGDH and PKM2 in vascular endothelium. It is possible that the AAV transfection targets the whole vascular tree, including that of the skeletal muscles. In fact, AAV-PHGDH and AAV-PKM2 mice have increased running distance, suggestive of an VO$_2$ max, which is the measurement of the maximum oxygen delivery and utilization for cardiovascular exercises. It has been reported that total plasma adiponectin concentrations are inversely correlated with the risk of coronary artery disease[61]. PPARγ agonists treatment induces the production of adiponectin to increase insulin sensitization and metabolic benefits[62]. Overexpression of PHGDH and PKM2 in mice increased the expression of key skeletal muscle proteins involved in metabolism and ROS control, which further confirms the improvement in muscle function and insulin sensitivity.

Estrogen is protective for cardiovascular function and estrogen is lost with aging[63]. Estrogen has been reported to reduce endothelial progenitor cell (EPC) senescence through augmentation of telomerase activity[64]. It is possible that estrogen may delay the senescence of endothelial cells in females. We compared the sex difference in our in vivo experiments. There was no marked difference for male and female mice on heart weight, heart rate for male and female (Fig. 7h, i). For LV ejection fraction, LV fractional shortening and exercise capability, overexpression of PHGDH and PKM2 displayed a little better beneficial effect in female mice than male mice (Fig. 7k–m; Supplementary Fig. 9f). This better effect in female mice may be related to the role of estrogen in delaying senescence.

There are several limitations of the current study. Firstly, for in vivo study, we overexpressed PHGDH and PKM2 in the endothelium. It remains unclear whether the transfection by AAV-PHGDH and AAV-PKM2 would affect the other wall cells considering that PKM2 can be exported outside of cells in the form of exosomes[65]. Secondly, it would be better to use the PKM2 knockout mouse and PHGDH knockout mouse to confirm the PHGDH-PKM2-H3pT11-SIRT1 axis in vivo. Meanwhile, it would be interesting to investigate the effect of serine on lifespan of WT, PKM2 knockout mice and PHGDH knockout mice.

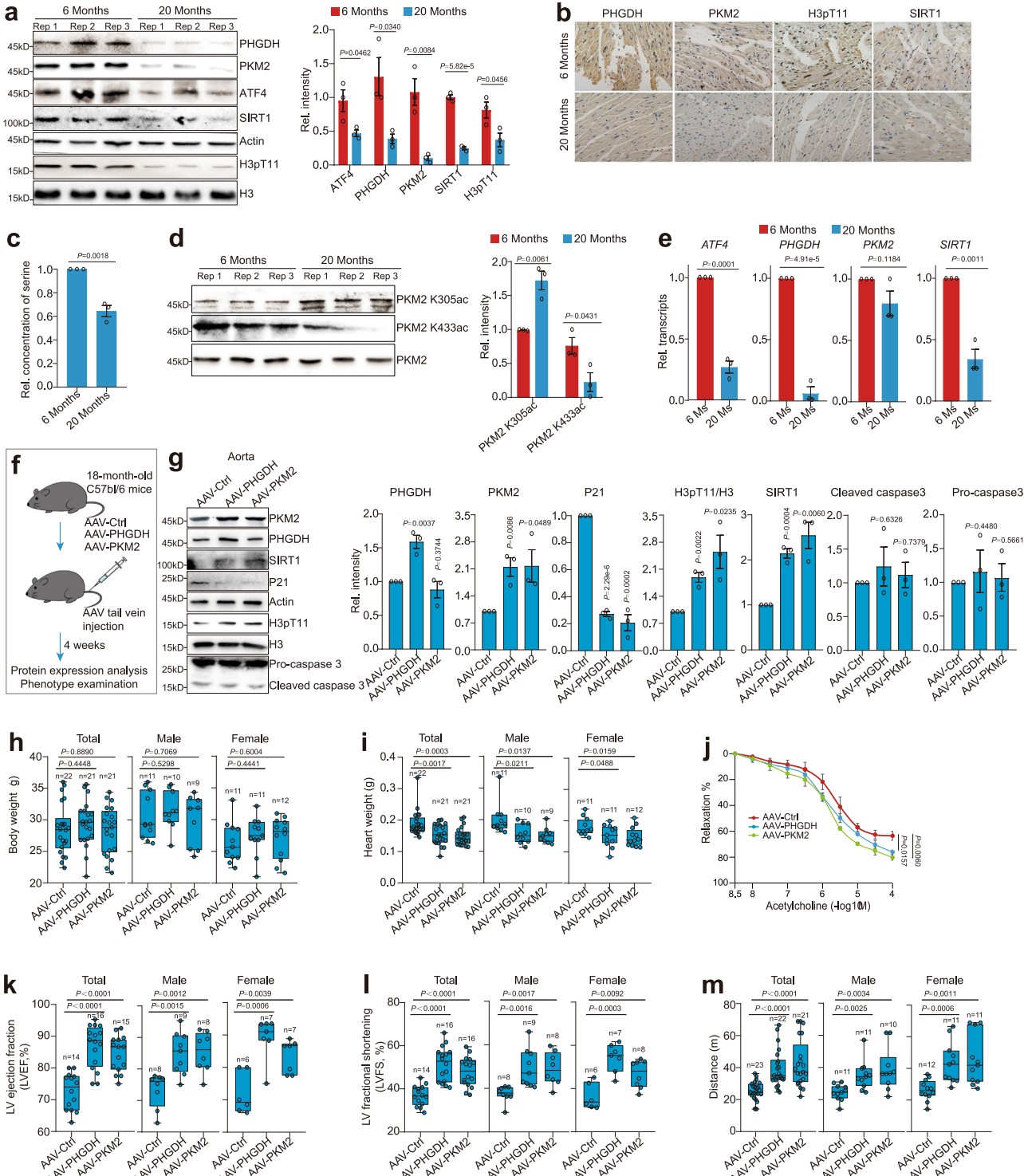

**Fig. 7 | PHGDH and PKM2 regulate aging phenotypes in mice. a** Immunoblots of ATF4, PHGDH, PKM2, SIRT1 and H3pT11 in the hearts of young mice (6 Months) and aged mice (20 Months). **b** Representative immunohistochemical staining of PHGDH, PKM2, H3pT11 and SIRT1 in the hearts of young mice (6 Months) and aged mice (20 Months). **c** Analysis of the relative serine concentrations in the hearts of young mice (6 Months) and aged mice (20 Months). **d** Immunoblots of PKM2 K305ac and PKM2 K433ac in the hearts of young mice (6 Months) and aged mice (20 Months). **e** RT-qPCR analysis of *ATF4*, *PHGDH*, *PKM2* and *SIRT1* transcription in the hearts of young mice (6 Months) and aged mice (20 Months). **f** 18-month-old C57BL/6 mice were injected with AAV that vascular endothelium (VE)-specific expression of PHGDH and PKM2 via tail vein. Four weeks later, protein expression was analyzed and the associated ageing phenotype was examined. **g** Immunoblots of PKM2, PHGDH, SIRT1, p21 and H3pT11 in the aorta of AAV-Ctrl, AAV-PHGDH and

AAV-PKM2 mice. Analysis of body weight (**h**) and heart weight (**i**) in AAV-Ctrl, AAV-PHGDH and AAV-PKM2 mice. **j** Concentration-response curves to acetylcholine obtained in aortic rings from AAV-Ctrl, AAV-PHGDH and AAV-PKM2 mice (4 mice per group). **k, l** Analysis of echocardiographic parameters, including left ventricular ejection fraction (LVEF), and left ventricular fractional shortening (LVFS) in AAV-Ctrl, AAV-PHGDH and AAV-PKM2 mice. **m** Total distance and time achieved by AAV-Ctrl, AAV-PHGDH and AAV-PKM2 mice in a treadmill running performance test. For **a, c-e, g, j**, data represent means ± SE; *n* = 3 independent experiments. For **h, i, k-m**, data represent means ± SE, the number of total mice, male mice and female mice were indicated in the graphs; centre lines denote medians, box limits denote 25th and 75th percentiles and whiskers denote maximum and minimum values. Two-sided *t*-tests were used for statistical analysis.

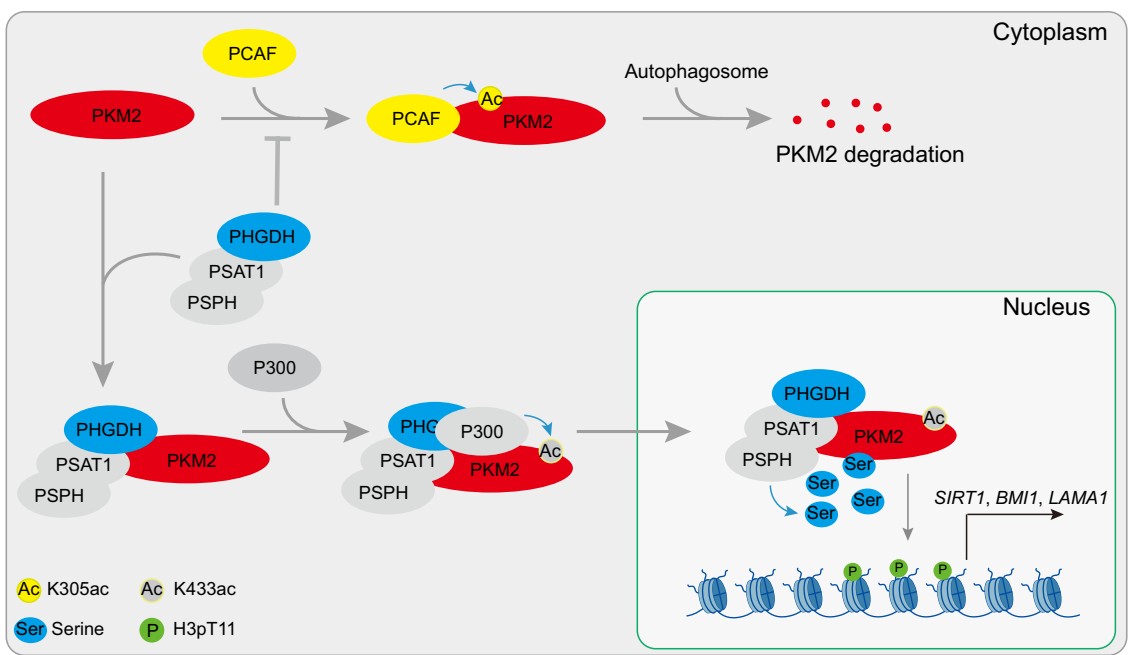

**Fig. 8 | Proposed model for regulation of senescence by the PHGDH-PKM2-H3pT11 axis.** PHGDH interacts with PKM2, which prevents PCAF-catalyzed PKM2 K305 and subsequent degradation by autophagy. PHGDH also enhances p300-catalyzed PKM2 K433 acetylation and promotes PKM2 nuclear translocation. PHGDH, together with PSAT1 and PSPH, provides the local serine to stimulate the activity of PKM2 to phosphorylate H3T11 and regulate the expression of senescence-associated genes. For simplicity, the multimerization status of proteins was not displayed in the cartoon.

Lastly, our study showed that over-expression of PKM2 and PHGDH increased the amount of collagen III, which may facilitate fibrosis. When PHGDH and PKM2 are needed to be over-expressed, it is better to prevent their potential side effect on glycine biosynthesis. One possible way is blocking the activity of SHMT1/2, which are the enzymes that convert serine to glycine.

In summary, we find that PHGDH-mediated serine synthesis plays an important role in delaying endothelial cell senescence. We uncover the PHGDH-PKM2-H3pT11 axis by which PHGDH regulates the expression of senescence-associated genes and prevents premature senescence. Therefore, our study shed lights on the functions of serine metabolism and glycolysis in cellular senescence and provide potential targets to ameliorate ageing phenotype.

## Methods

### Animals and treatments

Our research complies with all relevant ethical regulations. All procedures were approved by the Animal Care and Use Committee of Wuhan Sports University and Hubei University. Mice in a C57BL/6 background (male, female) were used in this study. For natural ageing experiments, 3-month-old mice in a C57BL/6 background (female) were purchased from Beijing Vital River Laboratory Animal Technology Co., Ltd. (Beijing, China). The mice were randomly divided into five groups (six mice per group) and housed under a 12 h light and 12 h dark cycle with an ambient temperature of $22 \pm 2\,°C$ and humidity of $55 \pm 10\%$ for up to 26 months. Water and standard chow were provided ad libitum following the regulations and guidelines of Wuhan Sports University and Hubei University. Mice in each group were killed, the heart, liver, kidney and spleen tissues were collected at 3, 6, 12, 20 and 26 months.

For adenovirus infection experiments, 18-month-old mice (male, female) in a C57BL/6 background were purchased from Beijing Vital River Laboratory Animal Technology Co., Ltd. (Beijing, China). The mice were housed under a 12 h light and 12-h dark cycle. Water and standard chow were provided ad libitum following the regulations and guidelines of Wuhan Sports University. The mice were divided into three groups (21–23 mice per groups). After one week of acclimatization, mice were then injected with 100 µl $5 \times 10^{11}$ viral genomes of AAV1-ICAM2-Control (AAV-Ctrl), AAV1-ICAM2-PHGDH (AAV-PHGDH) or AAV1-ICAM2-PKM2 (AAV-PKM2) particles through tail vein injection. AAV1-ICAM2-Control (AAV-Ctrl), AAV1-ICAM2-PHGDH (AAV-PHGDH) or AAV1-ICAM2-PKM2 (AAV-PKM2) particles were prepared and purchased from Taitool Bioscience (Shanghai, China). The mice used in this study were examined by the investigator in a double-blinded manner. One week later, the mice were injected with the same dose. The body weight of mice was measured every three days. After four weeks for injection, the ageing-related phenotypes were measured. The body weight of mice was measured every three days.

### Cell lines

The primary human umbilical vein endothelial cells (HUVECs) were purchased from ScienCell Research Laboratories. Cells were grown in Endothelial Cell Medium (ECM) supplemented with 5% of FBS, 1% of Endothelial Cell Growth Supplement (ECGs) and 1% of penicillin/streptomycin solution for up to 34 passages. The primary human aortic endothelial cells (HAECs) were purchased from Procell Life Science Technology (Wuhan). The serine free ECM was purchased from ScienCell. Most experiments for PHGDH-knockdown HUVECs were performed when cells were grown in serine-free medium. The senescent cells were verified by immunoblots with p21 (1:1000; 10355-1-AP, proteintech) and Senescence-associated β-galactosidase (SA-β-gal) staining. The HeLa and HEK293T cells were obtained from the American Type Culture Collection (ATCC). The HeLa and HEK293T cells were maintained in Dulbecco's modified Eagle's medium (DMEM) supplemented with 10% fetal bovine serum and 1% of penicillin/streptomycin solution. The cell lines used in this study were reauthenticated by short tandem repeat analysis after resuscitation in our laboratory. Normally, young passage HUVECs (P10) and senescent HUVECs (P20) were used for SAHF formation (DAPI staining) and SA-β-gal staining as indicated. Other experiments were performed with young passage HUVECs (P10).

## Plasmids and transfection

Expression plasmids for PHGDH, PKM2, SIRT1, H3 and SIRT2 were constructed by standard molecular biology techniques. Point mutations were generated by site-directed mutagenesis. For plasmid transfection, HeLa cells were cultured to 60–70% confluence and then transfected with 4 μg pCMV vector with reagent Neofect (TF20121201, Neofect). After 24 h, HeLa cells were collected for subsequent analysis.

Compared with HeLa cells, HUVECs were transfected with plasmids with lower efficiency. HUVECs were hence infected with lentiviruses to stably overexpress genes. In brief, HEK-293T cells were cultured to 70–80% confluence and then transfected with 2 μg pHAGE (expression vector), 1 μg psPAX2 and 1 μg pMD2.G plasmids with reagent Neofect (TF20121201, Neofect). After 48 h, the supernatant-containing lentivirus was collected. HUVECs were infected with lentivirus with 5 μg/ml polybrene by centrifuge at 3000 g for 1.5 h. After 24 h, HUVECs were selected using 1 μg/ml puromycin.

For siRNA transfection, the HUVECs or HeLa cells were cultured to 70–80% confluence and then transfected with 5 μl siRNA with reagent siRNA Mate (G04003, GenePharma). After 48 h, HUVECs and HeLa cells were collected for subsequent analysis. The knockdown efficiency was examined by immunoblots.

To construct stable knockdown HUVECs, shRNA hairpins were cloned into the lentiviral vector pLKO.1. A control hairpin that targeted GFP was cloned into the pLKO.1 vector and used as a negative control. HEK-293T cells were cultured to 70–80% confluence and then transfected with 2 μg pLKO.1 plasmid, 1 μg psPAX2 and 1 μg pMD2.G with reagent Neofect (TF20121201, Neofect). After 48 h, supernatant-containing lentivirus was collected. HUVECs were infected with lentivirus and selection was performed under 1 μg/ml puromycin. The siRNA sequences are following:

Scrambled siRNA, 5′-TTCTCCGAACGTGTCACGT-3′; PHGDH#1, GGGAACAGAGCUGAAUGGATT; PHGDH#2, CUGACCCUGUAGUA-CAGCATT; PKM2#1, AGGCAGAGGCUGCCAUCUATT; PKM2#2, CCAUAAUCGUCCUCACCAATT; PSAT1, UUCCAAGUUUGGUGU-GAUUTT; PSPH, CAAUAUCCCAGCAACCAAUTT; p300, UGACA-CAGGCAGGCUUGACTT; PCAF, GCAGAUACCAAACAAGUUUTT; LAMP2A, GAAAAUGCCACUUGCCUUUTT; PKM1/2, GGACCUGA-GAUCCGAACUGTT; ATF4, GCCUAGGUCUCUUAGAUGATT; Chk1, GCGUGCCGUAGACUGUCCATT.

## Senescence-associated β-galactosidase (SA-β-gal) staining

The SA-β-gal activity of HUVECs was measured by the Senescence β-Galactosidase Staining Kit (Beyotime)[14]. Cells were cultured in 6-well plates, washed with PBS and fixed for 10–15 min at room temperature. Cells were then washed twice with PBS and incubated with the staining mixture at 37 °C overnight. The SA-β-gal signals were analyzed using Image Pro Plus. For each staining assay, the late passage (P30) or $H_2O_2$-treated HUVECs were used as the positive control, while the early passage (P6) HUVECs were used as the negative control.

## Immunoblotting

Frozen tissue was homogenized in tissue lysis buffer (20 mM Tris-HCl pH7.5, 150 mM NaCl, 1 mM EDTA, 1% sodium deoxycholate, 1% Triton X-100) supplemented with protease inhibitor cocktails and phosphatase inhibitor cocktails. Cells were lysed with RIPA lysis buffer (Biosharp) containing 1 mM PMSF for 5 min on ice, sonicated for 1 min and boiled with SDS-sample buffer. The supernatants were subjected to 8–15% SDS-PAGE and transferred to a 0.2 μm polyvinylidene fluoride (PVDF, Bio-Rad) membrane. The membrane was blocked with 5% skim milk and incubated overnight with indicated primary antibodies in TBS supplemented with 0.1% Tween 20 at 4 °C. The membrane was then incubated with horseradish peroxidase (HRP)-labelled IgG secondary antibodies, developed with ECL Chemiluminescence Detection Kit (Biosharp) on a ChemiDoc Imaging System (Bio-Rad, USA). The band

intensity was quantified with ImageJ software (https://imagej.en.softonic.com/).

## Antibodies

Antibodies against H3 (1: 5000; ab1791), and H3pT11 (1:3000; ab5168) were purchased from Abcam; antibody against FLAG M2 (1:3000; F1804-1MG) was obtained from Sigma-Aldrich; antibodies against Myc (1:5000; 60003-2-1 g), PHGDH (1:3000; 14719-1-AP), GAPDH (1:10000; 10494-1-AP), PSAT1 (1:3000; 10501-1-AP), PSPH (1:3000; 14513-1-AP), PKM1 (1:2000; 15821-1-AP), p21 (1:3000; 10355-1-AP), PCAF (1:3000; 13983-1-AP), SIRT1 (1:2000; 13161-1-AP), 6×His (1:5000; HRP-66005), GST (1:5000; HRP-66001), beta-actin (1:10000; 20536-1-AP), and goat polyclonal anti-mouse IgG (1:5000; SA00001-1) were obtained from proteintech; antibodies against PKM2 (1:3000; 4053 S), p300 (1:5000; 54062), and ATF4 (1:1000; 11815) were purchased from Cell Signaling Technology; antibodies against mouse CDKN1A/p21 (1:3000; A11454), SOD1 (1:2000; A0274), SOD2 (1:1000; A19576), Caspase 3 (1:2000; A11040), PPARγ (1:1000; A11183), PGC1α (1:1000; A220995), and mouse PKM2 (1:5000; A19102) were purchased from Abclonal; antibody against SIRT6 (1:2000; 200499-6C9) was purchased from ZENBO; antibody against PKM2 K433ac (1:500) was custom-made in Abclonal. Antibody against PKM2K305ac was a gift from Dr. Qunying Lei (Fudan University). The specificity of the custom-made antibodies was confirmed by dot blots with peptides or immunoblots with cell extract of corresponding mutants.

## RNA sequencing (RNA-seq) and quantitative PCR analysis (RT-qPCR)

For replicative senescence, total RNA was extracted from different passages of HUVEC cells (P6, P20 and P30) by TRIzol reagent RNAiso Plus (Takara). Library construction, sequencing and bioinformatic analysis were done by Origingene Bio-pharm Technology Co. Ltd. (Shanghai). The P value is calculated by edgeR (v3.24). FDR (False Discovery Rate) is obtained by multiple testing and adjusting P value. The differentially expressed genes (DEGs) were defined as $P < 0.05$ and $log_2(FC) \geq 0.75$ or $log_2(FC) \leq -0.75$. DEGs were used for KEGG pathway analysis and KOBAS software was used to test the statistical enrichment of DEGs in KEGG pathways.

For RT-qPCR, RNA is extracted from cells with TRIzol reagent RNAiso Plus (Takara). Purified RNA was digested with DNase (Sigma) and reversed transcribed into cDNA using Reverse Transcriptase Kit (M-MLV) (ZOMANBIO). The cDNA was quantitated by qPCR with SYBR Green premix (Yeasen) using primers listed in Supplementary Table 1. The mRNA level of the gene of interest was normalized to that of *ACTIN*.

## Drug affinity responsive target stability (DARTS)

Cells were lysed in M-PER buffer (Thermo Scientific, 78501) with the addition of protease inhibitors (MCE, HY-K0010) and phosphatase inhibitors (Yeason, 20109). TNC buffer (50 mM Tris-HCl pH8.0, 50 mM NaCl, 10 mM $CaCl_2$) was added to the lysate and protein concentration was determined by the BCA Protein Assay kit (CWBIO, CW0014S). The cell lysate was then incubated with control or different concentrations of serine for 1 h on ice followed by 20 min incubation at room temperature. Digestion was performed using 50 μg/ml Pronase (Roche, 10165921001) at room temperature for 30 min and quenched using excess protease inhibitors. The lysate was prepared for mass spectrometry analysis by trypsin digestion as described[29]. For immunoblot analysis, 2×SDS loading buffer was added into the cell lysate after digestion followed by boiling for 5 min. Samples were subjected to SDS-PAGE and immunoblot analysis.

## Mass spectrometry analysis

Proteins were disulphide-reduced by 25 mM DTT at 37 °C for 40 min. Cysteines were alkylated by 50 mM iodoacetamide. Proteins were

digested with sequencing-grade trypsin (Promega) at 37 °C overnight and the supernatant was desalted using C18 solid-phase cartridges and lyophilized. The dried peptides were reconstituted in 0.1% FA and loaded onto an Acclaim PepMap 100 C18 LC column (Thermo Fisher) utilizing a Thermo Easy nLC 1000 LC system (Thermo Fisher) connected to Q Exactive HF mass spectrometer (Thermo Fisher) and analyzed[66].

### Immunoprecipitation (IP)

Cells were lysed in buffer A (50 mM Tris pH7.4, 150 mM NaCl, 1 mM EDTA, 6 mM sodium deoxycholate, 1% NP-40, 1 mM PMSF, 1:100 protease inhibitor cocktail) at 4 °C for 30 min. The supernatant was incubated overnight with indicated antibody conjugated Sepharose beads (GE Healthcare) at 4 °C, then washed with buffer A and boiled with 2× SDS-sample buffer for immunoblots.

### Chromatin immunoprecipitation (ChIP)

ChIP was performed as described with modifications[67]. Cells were cross-linked with 1% formaldehyde and quenched by 0.125 M glycine. Cells were collected, washed and lysed in lysis buffer B (50 mM Tris pH8.0, 5 mM EDTA, 1% SDS, 1 mM PMSF, 1:100 protease inhibitor cocktail). DNA was sheared by sonication and subjected to immunoprecipitation with antibodies pre-bound to Protein G Dynabeads (Invitrogen) overnight. Beads were washed and the eluted DNA/protein complexes were treated with 20 µg Proteinase K at 55 °C for 2 h and the crosslink was reversed at 65 °C overnight. The purified DNA was digested by RNase A and then quantitated by qPCR with specific primers listed in Supplementary Table 1.

### Bacterial expression and GST pull-down

The His-tagged or GST-tagged constructs were transformed into BL21 *E. coli* for protein expression. The transformed *E. coli* was induced with 1 mM IPTG at 16 °C overnight. Proteins were purified with Ni-NTA beads or glutathione beads and eluted with 100 mM imidazole or glutathione, respectively.

GST pull-down was performed according to standard protocols[14]. Briefly, the GST-tagged proteins were incubated with glutathione beads at 4 °C for 4 h. Pre-binding glutathione beads were then incubated with purified proteins in binding buffer C (50 mM Tris pH7.4, 150 mM NaCl, 1 mM EDTA, 6 mM sodium deoxycholate, 1% NP-40, 1 mM PMSF, protease inhibitor cocktail) at 4 °C for overnight. The beads were washed with buffer C and boiled with 2× SDS-sample buffer for immunoblots.

### Serine measurement

The concentration of serine was detected using ELISA kit (Bosk-Bio). Briefly, cells were lysed with RIPA lysis buffer. The cell lysate was incubated with HRP-labeled detection antibody in 96-well plates precoated with serine antibody at 37 °C for 1 h. The substrates TBM were then added and the absorbance at 450 nm was detected with microplate reader (SpectraMax M2, Molecular Devices).

### Immunofluorescence

Cells on the slides were fixed in 4% paraformaldehyde (Biosharp) and permeabilized with 0.2% Triton X-100 in PBS at room temperature for 15 min. Cells were incubated with primary antibodies at 4 °C overnight. Cells were then probed with Alexa Fluor-conjugated secondary antibody (Invitrogen), washed and subjected to 1 µg/ml 4,6-diamidino-2-phenylindole (DAPI) staining for 10 min.

### Subcellular fractionation

Cells were harvested, resuspended in buffer D (10 mM HEPES pH8.0, 10 mM KCl, 1.5 mM $MgCl_2$, 1 mM DTT, 1% Igepal CA-630, 1 mM PMSF, proteinase inhibitor cocktail). After centrifugation at 16,363 g for 10 min, the supernatant was collected as the cytoplasmic fraction. The pellet was resuspended in 80 µl RIPA lysis buffer (Biosharp) at 4 °C for 20 min. After sonicated for 1 min, the supernatant was collected as the nuclear fraction.

### In vitro kinase assay

The in vitro kinase assay was performed as described with modifications[68]. 200 ng purified PKM2 was incubated with 100 ng recombinant purified histone H3 in kinase buffer (100 mM HEPES pH7.4, 200 mM KCl, 12.4 mM MgOAc, 2 mM NaF, 5% glycerol, 0.15 mM PEP, 0.5 mM FBP, 1 mM PMSF, protease inhibitor cocktail, phosphatase inhibitor cocktail) in 40 µl at 25 °C for 0–1 h. The reaction was quenched by adding 2× SDS-sample buffer and boiling at 100 °C for 5 min. The reaction products were then subjected to immunoblots with anti-H3pT11 and anti-H3 antibodies.

### Tube formation assay

HUVECs ($1 \times 10^4$) were plated into 96-well plates precoated with 50 µl/well growth factor-reduced Matrigel (Corning, 354234) and incubated at 37 °C with 1% $O_2$ and 5% $CO_2$. After 12 h incubation, the tube counts and capillary lengths in each group were measured in five randomly chosen fields at ×100 magnification using ImageJ software (https://imagej.en.softonic.com/).

### Structure modeling

The crystal structure of PKM2 and p300 was obtained from the Protein Data Bank (PDB: 1ZJH, 4PZT). The structure of PHGDH was simulated through SWISS-MODEL (https://swissmodel.expasy.org). To visualize the docked conformation, the PyMol molecular graphics system was used, which also removed water molecules and peptide inhibitor PKI. The molecular docking simulation of PKM2, PHGDH and p300 was performed with the AutoDock Vina. All residues within PKM2 binding site were included using the following spatial coordinates of the central cavity: $x = 126$, $y = 126$, and $z = 126$. The coordinates of the grid resolution were $x = 34.469$, $y = 39.939$, $z = 34.724$.

### Treadmill performance test

The mice littermates were acclimated to the mouse wheel fatigue tester (Beijing Zhongshi Dichuang Technology Development Co., Ltd) for four consecutive days prior to the treadmill performance test. Acclimation consisted of 5 min of running at 6 m/min followed by 5 min of running at 10 m/min. For performance evaluation, mice were allowed to warm up for 3 min at 6 m/min. The speed was 12 m/min until the mice were fatigued when they no longer kept running. Treadmill electric current was set at 0.3 mA for both acclimation and testing sessions. Mice were weighed before the treadmill performance test was performed. The distance and time the mice ran was recorded.

### Echocardiographic imaging

Mice were anesthetized by intraperitoneal injection of 5% chloral hydrate before ultrasound imaging. The mice heart images were then taken with the rat and mouse ultrasound imaging system (Suzhou VINNO Technology Co., Ltd, VINNO6 Lab). For the echocardiographic imaging, the investigator lacked information regarding the treatment and was aware of only the strain and sex of the test mice.

### Blood pressure measurements

Intelligent non-invasive blood pressure system (Softron Biotechnology, BP-2010A) was used for assessing mice blood pressure. Mice were immobilized to prevent them from interfering with the assay during the formal experiment. The temperature of the bucket was set at 37 °C. After the mice calmed down, the blood pressure was measured and the results were recorded after the pulse wave stabilized. Each mouse was measured at least three times and the average was taken.

## Mice adiponectin measurement

Mice adiponectin levels were determined using the ELISA kit (Solarbio, SEKM-0142). Specifically, mouse blood was collected into anticoagulation tubes, rested for 20 min and centrifuged $1000 \times g$ for 10 min at 4 °C. The upper plasma layer was taken and diluted for measurement. The standard curve of adiponectin was obtained by measuring different concentrations of adiponectin standards.

## Endothelium-dependent relaxation assay

Thoracic aorta was isolated and placed in phosphate buffered solution (130 mM NaCl, 4.7 mM KCl, 1.18 mM $KH_2PO_4$, 1.17 mM $MgSO_4$, 24.9 mM $NaHCO_3$, 5.5 mM glucose, 0.026 mM EDTA and 1.6 mM $CaCl_2$). After separated from periadventitial fat and connective tissue, aortas were cut into ~2 mm rings. Each ring was positioned in a 5-ml organ bath of a DMT 620 M (Denmark). The organ bath was filled with PBS solution aerated continuously with a 95% $O_2$ and 5% $CO_2$ mixture and maintained at 37 °C. All vessels were allowed to equilibrate for at least 1 h at a resting tension of 4.5mN and stimulated with HI $K^+$ PBS (74.7 mM NaCl, 60 mM KCl, 1.18 mM $KH_2PO_4$, 1.17 mM $MgSO_4$, 24.9 mM $NaHCO_3$, 5.5 mM glucose, 0.026 mM EDTA, 1.6 mM $CaCl_2$) to verify its function. The vessels were constricted with 3 μM phenylephrine and then relaxed with increasing concentrations ($10^{-8.5}$ to $10^{-4}$ M) of acetylcholine. The tension was recorded and analyzed with LabChart software (v8).

## Statistics and reproducibility

Statistical differences in this study were determined by two-tailed unpaired $t$-test and a $P$ value<0.05 was considered statistically significant. All quantitative data were presented as mean values±SEM from at least three biological independent experiments. Prism 8 (v.8.0.1) was used to generate graphs.

## Reporting summary

Further information on research design is available in the Nature Portfolio Reporting Summary linked to this article.

## Data availability

All RNA-seq data in this study were deposited on GEO database. The GEO accession number for the raw RNA-seq data (young HUVECs and senescent HUVECs) in this manuscript is PRJNA764604. The GEO accession number for the raw RNA-seq data set (PKM2 knockdown, PHGDH knockdown, SIRT1 knockdown) in this manuscript is PRJNA673282 and PRJNA766438. The mass spectrometry proteomics data have been deposited to the ProteomeXchange Consortium via the PRIDE partner repository with the dataset identifier PXD035586 and PXD035585. The structure of PKM2 and p300 was obtained from the PDB (PDB: 1ZJH, 4PZT). Source data are provided with this paper.

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

## Acknowledgements

We would like to thank Dr. Li Wang (Fuwai Hospital, Chinese Academy of Medical Sciences) for assistance in endothelium-dependent relaxation assay. We also thank Dr. Qunying Lei (Fudan University) for providing us PKM2K305ac antibody, Dr. Wei-Guo Zhu (Shenzhen University) for providing us plasmids.

## Author contributions

Conceptualization: X.Y., S.L., T.Y. Experiments were performed by Y.W., L.T., H.H., Q.Y., B.H., G.W. Mass spec analysis was performed by F.G. Statistical analysis was performed by Y.W., T.Y., X.Y. Writing, review and editing: X.Y., S.L.

## Competing interests

The authors declare no competing interests.
