## [Peer Review File · Nature Communications]

Phosphoglycerate dehydrogenase activates PKM2 to phosphorylate histone H3T11 and attenuate cellular senescenceREVIEWER COMMENTS

Reviewer #1 (Remarks to the Author):

The authors reported that PHGDH is significantly reduced in ECs senescence. PHGDH prevents premature senescence primarily by enhancing the stability and activity of PKM2 through K305 and K433 acetylation. Although it lacks some in vivo data, for example, knockout mice and lifespan data, and most of the experiments are in vitro cellular work, the authors provided a detail molecular mechanism. Overall, this work reveals the molecular mechanism of the relationship between PHGDH and senescence; however, it is not a complete story and need to address some concerns.

1. Fig 7g showed blot of tissues from AAV injected mice. Over-expressing PHGDH or PKM2 upregulate SIRT1 and downregulate p21. However, why PKM2 did not upregulate in over-expressing PHGDH group? it is inconsistent with the authors theory that PHGDH-PKM2-SIRT1 axis in the anti-senescence mechanism. Besides, usually it needs at least three bio-replicated data of the mice tissues.
2. In Fig2g, over-expressing WT-PKM2 decrease the senescent markers. One would expect less or no effect when over-expressing kinase-dead mutation. However, why over-expressing K367M increase the senescent markers compared to control group?
3. In Fig4e, in the IP panel, there are more PKM2 when over-expressing PHGDH compared to control. However, in the input panel, there are also more PKM2 when over-expressing PHGDH. Therefore, it is hard to conclude that PHGDH enhanced the binding capacity between p300 and PKM2, because there are more PKM2 in the pool. This is the same question in Fig4f.
4. In Fig1e, P30 have less intracellular serine compared to P10. Could extra serine supplementation in the medium replenish the intracellular serine? In Fig1i and Fig suppl1i, serine supplementation in medium partly rescue senescent markers compared to serine free medium. However, In the serine containing normal medium, would extra serine supplementation rescue the senescent markers at P30 (Fig1d)?
5. The authors showed us that siPHGDH decrease PKM2 stability (Fig3e) and nuclear localization (Fig4a), to promote senescence. What is the effect to PKM2 if over-expressing PHGDH in senescent cells?
6. In Fig 3m, the authors directly expressed WT-PKM2 and K305R-PKM2 in HUVECs at P10. However, based on Fig 2d, there are still plenty of endogenous PKM2 in HUVECs at P10. Therefore, it may be hard to distinguish the effect of the endogenous and the exogenous PKM2.
7. The authors have showed us that PHGDH is downregulated in senescent HUVECS(Fig1d) and siPHGDH accelerate senescence of HUVECS(Fig1k). Would over-expressing PHGDH in senescent HUVECS rescue the senescence markers?
8. Some data need to be re-analyzed. In Fig1e,f,g,h,j. Fig2e, 4c, the relative value from control groups are all 1.0., leading to no SEM in control group and it would affect the P value. Usually, we define the mean value of control group to 1.0 and calculated the individual relative value in control group compared to mean value. Therefore, the control group should have SEM.

Reviewer #2 (Remarks to the Author):

The paper titled "Phosphoglycerate dehydrogenase activates PKM2 to phosphorylate histone H3T11 and attenuate cellular senescence" describes Phosphoglycerate dehydrogenase (PHGDH) regulation of Pyruvate Kinase M2 (PKM2) and their role in endothelial cell (EC) senescence using different in vitro and in vivo models. The results indicate PHGDH stabilizes PKM2, preventing autophagic degradation and promoting nuclear translocation via acetylation status of two different PKM2 lysine residues. In the nucleus PKM2 targets Histone H3T11, which controls senescence-associated genes such as SIRT1, BMI1, and LAMA1. Investigation into old vs. young mouse cardiac tissue revealed loss of these senescence protective markers. Finally, treatment with adenovirus containing either PHGDH or PKM2 rescued age induced cardiac dysfunction. Overall, the manuscript provides insightful information about novel players in endothelial cell senescence prevention.

Major.

- Figures 3J, 4E, and 4F do not have loading controls in experimental lanes. In order to make claims about PKM2/PCAF interactions with PHGDH in figure 3J, the actin loading control needs to be present. Similarly, panels 4E and 4F need actin loading controls in immunoprecipitated lanes with and without PHGDH oe and sh to make claims about PKM2/p300 interactions with PHGDH.
- Figure 7 shows in vivo over expression of PHGDH and PKM2, and the resulting cardiac enhancements. Figure 7H describes the changes in heart weight and body weight ratios. To ensure these changes are not due to increases in mouse weight and unchanged heart weight, please include the raw data for mouse heart and body weight separately.
- Panel 7B depicts histological staining of cardiac tissue from aged vs. young mice. Is a similar staining from AAV treated mice possible to corroborate echocardiography results?

Minor.

- In the results section please add limitations and future directions that are relevant to your study.
- Line 52 contains the phrase "can be fueled to synthesis" which should be changed to "fuels synthesis".
- Line 159 needs the word "is" placed between "which based".
- In line 202 please change "endogenous" to "endogenously".
- From the end of line 258 to the beginning of line 259 please remove "if not completely" as your data does not support the notion PHGDH completely inhibits PCAF.

Reviewer #3 (Remarks to the Author):

This is a very elegant study linking the serine metabolism pathway to senescence of endothelial cells, and I commend the authors for the experimental design: indeed, in cultured endothelial cells, it convincingly demonstrates that PHGDH prevents senescence by increasing the stability and activity of pyruvate kinase M2 that phosphorylates H3 at threonine 11 and thus influences gene expression.

My major concern is on the interpretation of the in vivo data.

First, they are not discussed, only to state that specific endothelial overexpression of PHGDH and PKM2 improved the cardiac ageing phenotype. The effects, both on cardiac function and running capacity, are indeed impressive considering that only endothelial cells have been transfected and that these effects are detected after a month only and from 18 m/o mice. Fig 7G, that is not discussed, shows that in the aorta, p21 decreases while SIRT1 and H3pT11 increase, supporting in vitro data; however, these represent WB of the whole aorta (same from suppl data on whole hearts, kidney and

liver). Assuming that endothelial cells represent likely 2% of the total cell number in the aorta, the expression of these proteins shown in Fig 7G must exceed the endothelial lineage and transfection must have influenced other wall cells. This should be validated because it is central to the hypothesis. These measures must be repeated after rubbing the endothelium from the lumen of the aorta: if endothelial only, signals should then be similar to AAV control.

The second impressive impact of the transfections is on exercise capacity. Understandably, the AAV transfection targets the whole vascular tree, including that of the skeletal muscles. Because both the running distance and the resistance are increased, suggestive of an increased VO₂max, further data are needed from the skeletal muscle to demonstrate that lower endothelial senescence improves blood delivery match and thus skeletal muscle performance, thus its molecular adaptation. At minima, expression of key skeletal muscle enzymes involved in ROS control and metabolism such as SOD1 and 2, PPARgamma and PCG1alpha, and, from the plasma, adiponectin, would confirm the improvement in muscle function and insulin sensitivity.

Third, the impact of the transfections on cardiac function is remarkable: the improved left ventricular function deserves more validation. It is essential to show histological cardiac structure and collagen content by staining. What is puzzling is the impact on heart rate. Why should HR be reduced after only a month? What could be the mechanism? What is the blood pressure in these mice? I assume that brain endothelial cells are also targeted and thus, the brain/heart axis must be affected also.

Finally, a true impact on endothelial function would be ideally to demonstrate a better endothelial flow-mediated dilation, or at least endothelium-dependent relaxation *ex vivo* in arteries.

The *in vivo* data are impressive and fascinating, but they need a direct validation that the effects seen are solely due to a reduction in the number of senescent endothelial cells as claimed; in addition, there is no evidence *in vivo* from the aorta data whether the transfection slowed senescence or had a senolytic effect (killed senescent cells).

Sex is not considered and it should be discussed; it could be a significant parameter considering that E2 is protective and may delay senescence of endothelial cells in females.

In any case, the overall data are impressive and the potential impact of the data in the field are high.

Reviewer #4 (Remarks to the Author):

In this paper, the authors suggested that serine is essential for attenuating the aging of endothelial cells. Although the role of serine in aging has been implicated previously, they revealed pyruvate kinase M2 (PKM2) as a target for serine and revealed that PKM2 interacts with phosphoglycerate dehydrogenase (PHGDH), a serine biosynthetic rate-limiting enzyme. They also elucidated the mechanism by which PHGDH regulates senescence-associated genes such as SIRT1 by increasing H3T11 phosphorylation through PKM2-K433 acetylation and nuclear translocation. The authors performed well-designed *in vitro* experiments to verify PHGDH-PKM2-H3pT11 axis and confirmed that PHGDH and PKM2 are involved in ameliorating the aging phenotype *in vivo*. However, following concerns should be addressed before final decision to publication.

Major Points

1. Previously, the relationship between serine and aging has already been indicated; through the studies that endogenous d-serine levels in the hippocampus decrease with aging [1] and that serine induces life span extension [2] etc. Therefore, it is better to explain the significance and novelty of this study compared to previous studies.

[1] Potier B, et al. (2010). *Front Aging Neurosci.* 2:1. [2] Edwards C, et al. (2015). *BMC Genet*, 16:8.

2. The authors used DARTS-LC-MS/MS method to investigate targets that resist degradation by pronase treatment due to binding of ligand small molecule, serine. However, there is no rationale for choosing 'PKM2' as the target of serine. Did the sequence coverage value of PKM2 belong to the top? Please provide in detail the criteria how target candidates were selected from detected proteins and provide the list of target candidates with high sequence coverage value.

3. In Fig 2b, the authors performed DARTS after expressing WT PKM2 or PKM2 H464A in cells and showed that PKM2 H464A mutant did not bind to serine supporting PKM2 could be a serine binding protein. However, the PKM2 antibody cannot differentiate between endogenous PKM2 and expressed WT PKM2, PKM2 H464A. If the expression vector includes the tag (FLAG or Myc ?), it would be better to observe the DARTS pattern for ectopic PKM2 using FLAG or Myc antibodies instead. In addition, it would be better to investigate the binding activity of PKM2 H464A (serine non-binding mutant) with PHGDH and its phosphorylation activity to H3T11 in the regulation of senescence-associated genes (SIRT1, BMI1, LAMA1) regulation.

Minor points

1. Add information about the number of passages of HUVECs, pronase and serine concentration, pronase treated time for DARTS experiments.
2. [Figure 2a] In CBB-stained gel images, it is difficult to determine whether the indicated band is exactly PKM2 or other proteins of similar size.
3. [Figure 2b] The authors used two serine concentrations, and no concentration dependence was observed. Please add possible discussion for this.

Point-by-point response to the referees' comments

REVIEWER COMMENTS

Our responses are in blue.

Reviewer #1 (Remarks to the Author):

The authors reported that PHGDH is significantly reduced in ECs senescence. PHGDH prevents premature senescence primarily by enhancing the stability and activity of PKM2 through K305 and K433 acetylation. Although it lacks some in vivo data, for example, knockout mice and lifespan data, and most of the experiments are in vitro cellular work, the authors provided a detail molecular mechanism. Overall, this work reveals the molecular mechanism of the relationship between PHGDH and senescence; however, it is not a complete story and need to address some concerns.

We would like to thank this reviewer for constructive comments and suggestions. In the revised manuscript, we have addressed all comments raised by the reviewer and revised the manuscript in light of the reviewer's concerns. The point-by-point answers to all the issues raised by the reviewer are enumerated below.

1. Fig. 7g showed blot of tissues from AAV injected mice. Over-expressing PHGDH or PKM2 upregulate SIRT1 and downregulate p21. However, why PKM2 did not upregulate in over-expressing PHGDH group? it is inconsistent with the authors theory that PHGDH-PKM2-SIRT1 axis in the anti-senescence mechanism. Besides, usually it needs at least three bio-replicated data of the mice tissues.

We thank the reviewer for this point. In the original Fig. 7g, we used the anti-FLAG antibody, which can only detect the ectopically expressed FLAG-PKM2 but not endogenously expressed PKM2. That is the reason why PKM2 was not up-regulated in AAV-PHGDH mice. In the revised manuscript, we re-performed the immunoblots with anti-PKM2 for three bio-replicated mice tissues and quantified the results. Our data showed that the expression of PKM2 was significantly increased in AAV-PHGDH mice (Fig. 7g), which is consistent with our conclusion that the PHGDH-PKM2-H3pT11 axis was enhanced in the aortae of AAV-PHGDH and AAV-PKM2 mice.

2. In Fig. 2g, over-expressing WT-PKM2 decrease the senescent markers. One would expect less or no effect when over-expressing kinase-dead mutation. However, why over-expressing K367M increase the senescent markers compared to control group?

We thank the reviewer for this point. As PKM2 has been reported to exist in a dimer or tetramer state inside cells (*Mol Cell* 2012, 45(5):598-609), it is possible that over-expression of PKM2 kinase-dead mutation (PKM2 K367M) may interfere with the

function of the endogenous PKM2, leading to a slightly increased effect on senescence markers (**Fig. 2g**). To exclude this effect from the endogenous PKM2, we used PKM2 siRNA to knock down the endogenous PKM2 in HUVECs that stably overexpress siRNA resistant WT PKM2 and PKM2 K367M. Knockdown of PKM2 accelerated cellular senescence and over-expression of WT PKM2 but not kinase-dead PKM2 K367M attenuated cellular senescence (**Fig. 2g**). We have put this data in the revised manuscript (**Fig. 2g**).

As senescent HUVECs (P20) have fewer endogenous PKM2 (**Fig. 2d**), we also overexpressed WT PKM2 and PKM2 K367M in HUVECs (P20). The SA- β -gal staining assay showed that over-expression of WT PKM2 significantly reduced cellular senescence; however, there was no significant difference between control (Ctrl) and PKM2 K367M (**Rebuttal Letter Fig. 1**).

Rebuttal Letter Fig. 1. Effect of over-expression of WT PKM2 and its mutants (PKM2 K367M, H464A) on cellular senescence as determined by SA- β -gal staining. HUVECs (P20) were cultured in serine-containing medium and then infected with lentiviruses to stably overexpress PKM2 (WT PKM2, K367M, H464A). Right panel: Quantification of the number of SA- β -gal positive HUVECs.

Page 48, paragraph 1: Changed “HUVECs were cultured in serine-containing medium and infected with lentiviruses to stably overexpress PKM2 (WT PKM2, K367M, H464A).” to “HUVECs that stably overexpress PKM2 (WT PKM2, K367M, H464A) were transfected with siPKM2 to knock down the endogenous PKM2. The ectopic expressed PKM2 was resistant to siPKM2.”

3. In Fig. 4e, in the IP panel, there are more PKM2 when over-expressing PHGDH compared to control. However, in the input panel, there are also more PKM2 when over-expressing PHGDH. Therefore, it is hard to conclude that PHGDH enhanced the binding capacity between p300 and PKM2, because there are more PKM2 in the pool. This is the same question in Fig. 4f.

We thank the reviewer for this point. In the input panel of original Fig. 4e, it is true that there is more PKM2 in PHGDH-overexpression (PHGDH OE) cells, which is consistent with our conclusion that PHGDH promotes PKM2 stability. In the revised manuscript, to show that over-expression of PHGDH enhanced the interaction between PKM2 and p300, we re-performed the Co-IP by using less PHGDH OE cell lysates for immunoprecipitation to ensure the equal amount of PKM2 in the input, which was indicated by less actin and p300 in the input samples (**Fig. 4e**). Under this condition, we still observed more PKM2 Co-IPed with p300 in PHGDH OE cells (**Fig. 4e**).

For Fig. 4f, we also re-performed the Co-IP by using more cell lysates for PHGDH-knockdown (shPHGDH) cells to ensure the equal amount of PKM2 in the input, which was indicated by more actin and p300 in the input samples (**Fig. 4f**). We can clearly observe less PKM2 was IPed by p300 in shPHGDH cells (**Fig. 4f**). We've put these results in the revised manuscript.

In addition to these in vivo Co-IP, our in vitro Co-IP data with recombinant purified PHGDH and PKM2 also showed that PHGDH directly promoted the interaction between p300 and PKM2 (**Fig. 4g**). Therefore, both in vivo and in vitro Co-IP data support the conclusion that PHGDH enhances the binding capacity between p300 and PKM2.

Page 50, paragraph 2: Added “To ensure equal loading of PKM2, less cell lysate from PHGDH-overexpression (PHGDH OE) HUVECs (**e**) and more lysate from PHGDH-knockdown (shPHGDH) HUVECs (**f**) were used for immunoprecipitation.”

4. In Fig. 1e, P30 have less intracellular serine compared to P10. Could extra serine supplementation in the medium replenish the intracellular serine? In Fig1i and Fig suppli, serine supplementation in medium partly rescue senescent markers compared to serine free medium. However, In the serine containing normal medium, would extra serine supplementation rescue the senescent markers at P30 (Fig. 1d)?

We thank this reviewer for this comment. It is true that extra serine supplementation in the medium can replenish the senescence-caused reduction of intracellular serine (**Rebuttal Letter Fig. 2a**). Extra serine supplementation significantly reduced cellular senescence in P10 HUVECs when cultured in serine-containing medium. Extra serine supplementation slightly reduced cellular senescence in P30 HUVECs (**Rebuttal Letter Fig. 2b**). As one major target of serine is PKM2 in this study, we thus examined PKM2 in P30 HUVECs. Very few PKM2 in P30 HUVECs can be detected (**Rebuttal Letter Fig. 2c**), which is consistent with our data that PKM2 is degraded by autophagy during cellular senescence (**Fig. 2d**). The fewer PKM2 in P30 HUVECs may account for this mild effect of serine on cellular senescence.

Rebuttal Letter Fig. 2. Effect of exogenous serine on cellular senescence in different passages of HUVECs. HUVECs (P10 and P30) were cultured in serine-containing normal medium and then treated with 1 mM serine for 24 hr. The relative intracellular serine levels were measured (a). HUVECs senescence was determined by SAHF formation (DAPI staining) and SA-β-gal staining (b). The intracellular PKM2 protein levels were examined by immunoblots (c).

5. The authors showed us that siPHGDH decrease PKM2 stability (Fig. 3e) and nuclear localization (Fig. 4a), to promote senescence. What is the effect to PKM2 if over-expressing PHGDH in senescent cells?

That is a good point. We overexpressed PHGDH in senescent HUVECs (P20) and observed increased expression of PKM2 (Fig. 3d). Moreover, overexpression of PHGDH also promoted the nucleus translocation of PKM2 in HUVECs (P20) (Supplementary Fig. 4I). We've put these results in the revised manuscript (Fig. 3d; Supplementary Fig. 4I).

Page 13, paragraph 1: Added "Overexpression of PHGDH promoted the nuclear translocation of PKM2 even in replicative senescent (P20) HUVECs (Supplementary Fig. 4I)."

Page 49, paragraph 1: Changed "HUVECs were infected with lentiviruses to stably knock down PHGDH (shPHGDH) or overexpress PHGDH (PHGDH OE)." to "HUVECs (P10) were infected with lentiviruses to stably knock down PHGDH (shPHGDH). HUVECs (P10, young; P20, senescent) were infected with lentiviruses to overexpress PHGDH (PHGDH OE)."

6. In Fig 3m, the authors directly expressed WT-PKM2 and K305R-PKM2 in HUVECs at P10. However, based on Fig 2d, there are still plenty of endogenous PKM2 in HUVECs at P10. Therefore, it may be hard to distinguish the effect of the endogenous and the exogenous PKM2.

We thank this reviewer for this comment. To distinguish the effect of endogenous and

the exogenous PKM2, we used PKM2 siRNA to knock down the endogenous PKM2 in HUVECs that stably overexpress siRNA resistant WT PKM2 and PKM2 K305R. Compared with WT PKM2, PKM2 K305R mutation increased PKM2 and substantially attenuated cellular senescence (**Fig. 3m**). We've put these results in the revised manuscript (**Fig. 3m**).

Page 12, paragraph 2: Changed “We then examined the effect of PKM2 K305ac on cellular senescence by analyzing the senescence phenotype of HUVECs when infected with lentiviruses to stably overexpress WT PKM2 and PKM2 K305R (to mimic deacetyl modification).” to “We then examined the effect of PKM2 K305ac on cellular senescence. HUVECs that stably overexpress WT PKM2 and PKM2 K305R (to mimic deacetyl modification) were transfected with siPKM2 to knock down the endogenous PKM2.”

Page 50, paragraph 1: Changed “HUVECs were infected with lentiviruses to stably express FLAG-WT PKM2 and FLAG-PKM2 K305R.” to “HUVECs that stably express FLAG-WT PKM2 and FLAG-PKM2 K305R were transfected with siPKM2 to knock down the endogenous PKM2. The ectopically expressed PKM2 was resistant to siPKM2.”

7. The authors have showed us that PHGDH is downregulated in senescent HUVECS (Fig. 1d) and siPHGDH accelerate senescence of HUVECS (Fig. 1k). Would over-expressing PHGDH in senescent HUVECS rescue the senescence markers?

We thank this reviewer for this comment. We overexpressed PHGDH in senescent HUVECs (P20) and then examined the senescence markers. SA- β -gal staining and SAHF assays revealed delayed cellular senescence in PHGDH-overexpression (PHGDH OE) HUVECs (**Supplementary Fig. 2d**). Overexpression of PHGDH also rescued the expression of *P21* and *HMGB1* in P20 HUVECs (**Supplementary Fig. 2e**). All these data indicate that overexpression of PHGDH can rescue the senescence markers in senescent HUVECs. This effect may be related to the fact that overexpression of PHGDH can increase the protein level of PKM2 and its nuclear translocation in senescent cells (**Fig. 3d; Supplementary Fig. 4l**).

Page 6, paragraph 2: Changed “delayed cellular senescence in PHGDH overexpression (PHGDH OE) HUVECs” to “Overexpression of PHGDH (PHGDH OE) delayed cellular senescence in both young (P10) and replicative senescent (P20) HUVECs (Supplementary Fig. 2d, e).”

Page 23, paragraph 2: Added “Overexpression of PHGDH can increase the protein level of PKM2, promote its nuclear translocation and attenuate cellular senescence in both young and senescent HUVECs.”

8. Some data need to be re-analyzed. In Fig1e, f, g, h, j. Fig2e, 4c, the relative value from control groups are all 1.0., leading to no SEM in control group and it would affect the P value. Usually, we define the mean value of control group to 1.0 and calculated the individual relative value in control group compared to mean value. Therefore, the control group should have SEM.

We thank this reviewer for this comment. We made changes to Fig. 1e, 1f, 1g, 1h, 1j, 2e, 4c as requested.

Reviewer #2 (Remarks to the Author):

The paper titled “Phosphoglycerate dehydrogenase activates PKM2 to phosphorylate histone H3T11 and attenuate cellular senescence” describes Phosphoglycerate dehydrogenase (PHGDH) regulation of Pyruvate Kinase M2 (PKM2) and their role in endothelial cell (EC) senescence using different in vitro and in vivo models. The results indicate PHGDH stabilizes PKM2, preventing autophagic degradation and promoting nuclear translocation via acetylation status of two different PKM2 lysine residues. In the nucleus PKM2 targets Histone H3T11, which controls senescence-associated genes such as SIRT1, BMI1, and LAMA1. Investigation into old vs. young mouse cardiac tissue revealed loss of these senescence protective markers. Finally, treatment with adenovirus containing either PHGDH or PKM2 rescued age induced cardiac dysfunction. Overall, the manuscript provides insightful information about novel players in endothelial cell senescence prevention.

We would like to thank this reviewer for recognition of our work. We also thank this reviewer for constructive comments and suggestions. In the revised manuscript, we have addressed the three major comments and other minor comments raised by the reviewer and revised the manuscript accordingly. The point-by-point answers to all the issues raised by the reviewer are enumerated below.

Major.

1. Figures 3J, 4E, and 4F do not have loading controls in experimental lanes. In order to make claims about PKM2/PCAF interactions with PHGDH in figure 3J, the actin loading control needs to be present. Similarly, panels 4E and 4F need actin loading controls in immunoprecipitated lanes with and without PHGDH oe and sh to make claims about PKM2/p300 interactions with PHGDH.

We thank this reviewer for this comment. For Fig. 3j, we examined the amount of actin

in the input and IP samples. We observed equal amount of actin and PCAF in the input samples but no actin was observed in the IP samples, which could be due to no interaction between PKM2 and actin (**Fig. 3j**). We thus examined the presence of STAT3, which has been reported to be phosphorylated by PKM2 (*Mol Cell* 2012, 45, 598-609). Our data showed that the amount of STAT3 was similar in the input and IP samples for control and PHGDH OE (**Fig. 3j**).

For Fig. 4e, we re-performed the Co-IP by using less PHGDH OE cell lysates for immunoprecipitation to ensure the equal amount of PKM2 in the input, which was indicated by less actin and p300 in the input samples (**Fig. 4e**). As actin was not observed in the IP samples, we examined histone H3, which is a substrate for p300. Histone H3 was less in PHGDH OE input and IP samples (**Fig. 4e**), suggesting that the interaction between p300 and histone H3 is unaffected by PHGDH over-expression. When less cell lysate was used for IP, more PKM2 was still Co-IPed with p300 in PHGDH OE cells (**Fig. 4e**), suggesting that over-expression of PHGDH promotes PKM2 and p300 interaction.

For Fig. 4f, we re-performed the Co-IP by using more cell lysates for PHGDH-knockdown (shPHGDH) cells to ensure the equal amount of PKM2 in the input, which was indicated by more actin, p300 and H3 in the input samples (**Fig. 4f**). Histone H3 was higher in shPHGDH input and IP samples (**Fig. 4f**). When more cell lysate was used for IP, less PKM2 was IPed by p300 in shPHGDH cells (**Fig. 4f**).

Page 49, paragraph 1: Added “Actin and STAT3 were used as loading controls.”

Page 51, paragraph 1: Added “To ensure equal loading of PKM2, less cell lysate from PHGDH-overexpression (PHGDH OE) HUVECs (**e**) and more lysate from PHGDH-knockdown (shPHGDH OE) HUVECs (**f**) were used for immunoprecipitation. Actin and histone H3 were used as loading controls.”

2. Figure 7 shows in vivo over expression of PHGDH and PKM2, and the resulting cardiac enhancements. Figure 7H describes the changes in heart weight and body weight ratios. To ensure these changes are not due to increases in mouse weight and unchanged heart weight, please include the raw data for mouse heart and body weight separately.

We thank this reviewer for this comment. We put the raw data for mouse heart weight and body weight separately in the revised manuscript (**Fig. 7h, i**). Our data showed that the body weight was not significantly changed in in AAV-PHGDH and AAV-PKM2 mice (**Fig. 7h**). The heart weight was significantly reduced in AAV-PHGDH and AAV-PKM2 mice (**Fig. 7i**). Thus, the decrease of the heart weight and body weight ratio was due to reduced heart weight but not due to increases in mouse body weight.

Page 21, paragraph 1: Changed “Although VE-specific expression of PHGDH and PKM2 had no significant effect on body weight (Supplementary Fig. 8d), the heart/body weight ratio was significantly decreased by VE-targeted expression of PHGDH and PKM2 (Fig. 7h),” to “Although VE-specific expression of PHGDH and PKM2 had no significant effect on body weight (Fig. 7h), the heart weight and the heart weight/body weight ratio were significantly decreased by VE-targeted expression of PHGDH and PKM2 (Fig. 7i; Supplementary Fig. 9a)”.

3. Panel 7B depicts histological staining of cardiac tissue from aged vs. young mice. Is a similar staining from AAV treated mice possible to corroborate echocardiography results?

We performed histological staining of cardiac tissue from AAV-Ctrl, AAV-PHGDH and AAV-PKM2 mice in the revised manuscript. The amount of PKM2, H3pT11 and SIRT1 was increased in AAV-PHGDH and AAV-PKM2 mice (**Supplementary Fig. 8f**). The amount of PHGDH was only increased in AAV-PHGDH (**Supplementary Fig. 8f**). All these histological staining data are consistent with our immunoblot data (**Fig. 7g**)

Page 20, paragraph 1: Changed “With the VE-specific expression of PHGDH and PKM2, H3pT11 and SIRT1 were also significantly increased (Fig. 7g).” to “With the VE-specific expression of PHGDH and PKM2, H3pT11 and SIRT1 were also significantly increased as indicated by immunoblots and histological staining of cardiac tissue (Fig. 7g; Supplementary Fig. 8f).”

Minor.

4. In the results section please add limitations and future directions that are relevant to your study.

The following are limitations and future directions for our study. Firstly, for in vivo study, we overexpressed PHGDH and PKM2 in the endothelium. It remains unclear whether transfection by AAV-PHGDH and AAV-PKM2 would affect the other wall cells, especially considering that PKM2 can be exported outside of cells in the form of exosomes (*J Proteomics* 2010, 73, 1907-20). Secondly, it would be better to use the PKM2 knockout mouse and PHGDH knockout mouse to confirm the PHGDH-PKM2-H3PT11-SIRT1 axis in vivo. Meanwhile, it would be interesting to investigate the effect of serine on lifespan of WT, PKM2 knockout mice and PHGDH knockout mice.

Page 28, paragraph 3: Added “There are several limitations of the current study. There are several limitations of the current study. Firstly, for in vivo study, we overexpressed PHGDH and PKM2 in the endothelium. It remains unclear whether the transfection by AAV-PHGDH and AAV-PKM2 would affect the other wall cells considering that PKM2 can be exported outside of cells in the form of exosomes⁵⁷. Secondly, it would be better

to use the PKM2 knockout mouse and PHGDH knockout mouse to confirm the PHGDH-PKM2-H3PT11-SIRT1 axis in vivo. Meanwhile, it would be interesting to investigate the effect of serine on lifespan of WT, PKM2 knockout mice and PHGDH knockout mice.”

5. Line 52 contains the phrase “can be fueled to synthesis” which should be changed to “fuels synthesis”.

We made changes as requested.

Page 3, paragraph 1: Changed ““can be fueled to synthesis”” to “fuels synthesis”.

6. Line 159 needs the word “is” placed between “which based”.

We made changes as requested.

Page 7, paragraph 3: Changed “which based on” to “which is based on”.

7. In line 202 please change “endogenous” to “endogenously”.

We made changes as requested.

Page 9, paragraph 2: Changed “between endogenous expressed” to “between endogenously expressed”.

8. From the end of line 258 to the beginning of line 259 please remove “if not completely” as your data does not support the notion PHGDH completely inhibits PCAF.

We made changes as requested.

Page 12, paragraph 2: Changed “partly, if not completely by inhibiting” to “partly by inhibiting”.

Reviewer #3 (Remarks to the Author):

This is a very elegant study linking the serine metabolism pathway to senescence of endothelial cells, and I commend the authors for the experimental design: indeed, in cultured endothelial cells, it convincingly demonstrates that PHGDH prevents

senescence by increasing the stability and activity of pyruvate kinase M2 that phosphorylates H3 at threonine 11 and thus influences gene expression.

We sincerely thank this reviewer for recognition of our work. We also thank this reviewer for the constructive suggestions and comments, which have significantly improved the quality of this manuscript. In the revised manuscript, we re-performed some *in vivo* experiments and tried our best to interpret these *in vivo* data.

My major concern is on the interpretation of the *in vivo* data.

1. First, they are not discussed, only to state that specific endothelial overexpression of PHGDH and PKM2 improved the cardiac ageing phenotype. The effects, both on cardiac function and running capacity, are indeed impressive considering that only endothelial cells have been transfected and that these effects are detected after a month only and from 18 m/o mice. Fig 7G, that is not discussed, shows that in the aorta, p21 decreases while SIRT1 and H3pT11 increase, supporting *in vitro* data; however, these represent WB of the whole aorta (same from suppl data on whole hearts, kidney and liver). Assuming that endothelial cells represent likely 2% of the total cell number in the aorta, the expression of these proteins shown in Fig 7G must exceed the endothelial lineage and transfection must have influenced other wall cells. This should be validated because it is central to the hypothesis. These measures must be repeated after rubbing the endothelium from the lumen of the aorta: if endothelial only, signals should then be similar to AAV control.

We thank this reviewer for this point. To determine whether AAV-PHGDH and AAV-PKM2 were specifically expressed in endothelial cells, we re-performed the *in vivo* experiments according to the method suggested by the reviewer. We took the aortae from AAV-Ctrl, AAV-PHGDH and AAV-PKM2 mice and then rubbed the endothelium from the aorta lumen. The level for PKM2, H3pT11 and SIRT1 was significantly increased in the endothelium of the AAV-PHGDH and AAV-PKM2 mice (**Supplementary Fig. 8d**), which is consistent with our *in vitro* data. After the endothelium was removed, the remainder aortae from AAV-PHGDH and AAV-PKM2 mice had similar expression of PKM2, H3pT11 and SIRT1 with AAV-Ctrl (**Supplementary Fig. 8d**), confirming the endothelium-specific expression of PHGDH and PKM2. As PKM2 can be exported outside of cells in the form of exosomes (*J Proteomics* 2010, 73(10):1907-20), we cannot exclude the possibility that transfection of AAV-PHGDH and AAV-PKM2 may affect the expression of proteins in other wall cells. We have added this in the discussion section.

Page 20, paragraph 1: Added “The expression of FLAG-PHGDH and FLAG-PKM2 was detected in the endothelium and aortae in AAV-PHGDH and AAV-PKM2 mice, respectively (Fig. 7g; Supplementary Fig. 8d). After the endothelium was rubbed from the aortae of AAV-PHGDH and AAV-PKM2 mice, FLAG-PHGDH and FLAG-PKM2

cannot be detected (Fig. 7g; Supplementary Fig. 8d), suggesting the endothelium-specific expression of PHGDH and PKM2. Moreover, the protein levels of PKM2 were increased in the aorta endothelium of AAV-PHGDH mice but not in the endothelium-rubbed aortae (Supplementary Fig. 8d).”

Page 28, paragraph 3: Added “Firstly, for in vivo study, we overexpressed PHGDH and PKM2 in the endothelium. It remains unclear whether the transfection by AAV-PHGDH and AAV-PKM2 would affect the other wall cells considering that PKM2 can be exported outside of cells in the form of exosomes⁶⁴.”

2. The second impressive impact of the transfections is on exercise capacity. Understandably, the AAV transfection targets the whole vascular tree, including that of the skeletal muscles. Because both the running distance and the resistance are increased, suggestive of an increased VO₂max, further data are needed from the skeletal muscle to demonstrate that lower endothelial senescence improves blood delivery match and thus skeletal muscle performance, thus its molecular adaptation. At minima, expression of key skeletal muscle enzymes involved in ROS control and metabolism such as SOD1 and 2, PPAR γ and PGC1 α , and, from the plasma, adiponectin, would confirm the improvement in muscle function and insulin sensitivity.

We thank this reviewer for this comment. We agree with the reviewer that the AAV transfection may target the whole vascular tree, including that of the skeletal muscles. The AAV-PHGDH and AAV-PKM2 mice have increased running distance, suggestive of an VO₂ max. We thus examined the expression of key skeletal muscle enzymes involved in ROS control and metabolism, including SOD1, SOD2, PPAR γ and PGC1 α . Overexpression of PHGDH and PKM2 up-regulated the expression of key skeletal muscle proteins involved in metabolism, including peroxisome proliferator-activated receptor- γ (PPAR γ) and its coactivator PGC-1 α as specific markers of mitochondrial biogenesis (**Supplementary Fig. 9g, h**). The expression of adiponectin was significantly increased in the plasma from AAV-PHGDH and AAV-PKM2 mice (**Supplementary Fig. 9i**). The enzymes involved in ROS control, including SOD1 and SOD2 were also up-regulated in AAV-PHGDH and AAV-PKM2 mice (**Supplementary Fig. 9g, h**). These data further confirmed the improvement in muscle function. We have added these data in the revised manuscript (**Supplementary Fig. 9**).

Page 21, paragraph 2: Added “VE-targeted expression of PHGDH and PKM2 up-regulated the expression of key skeletal muscle proteins involved in metabolism, including peroxisome proliferator-activated receptor- γ (PPAR γ), PPAR γ coactivator PGC-1 α as specific markers of mitochondrial biogenesis (Supplementary Fig. 9g, h). The levels of plasma adiponectin were also significantly increased in AAV-PHGDH and AAV-PKM2 mice (Supplementary Fig. 9i). Moreover, the enzymes involved in ROS control, including SOD1 and SOD2 were also up-regulated in AAV-PHGDH and AAV-PKM2 mice (Supplementary Fig. 9g, h).”

Page 28, paragraph 1: Added “We used a recombinant adenovirus-associated virus to express PHGDH and PKM2 in vascular endothelium. It is possible that the AAV transfection targets the whole vascular tree, including that of the skeletal muscles. In fact, AAV-PHGDH and AAV-PKM2 mice have increased running distance, suggestive of an VO₂ max, which is the measurement of the maximum oxygen delivery and utilization for cardiovascular exercises. It has been reported that total plasma adiponectin concentrations are inversely correlated with the risk of coronary artery disease⁶⁰. PPAR γ agonists treatment induces the production of adiponectin to increase insulin sensitization and metabolic benefits⁶¹. Overexpression of PHGDH and PKM2 in mice increased the expression of key skeletal muscle proteins involved in metabolism and ROS control, which further confirms the improvement in muscle function and insulin sensitivity.”

3. Third, the impact of the transfections on cardiac function is remarkable: the improved left ventricular function deserves more validation. It is essential to show histological cardiac structure and collagen content by staining. What is puzzling is the impact on heart rate. Why should HR be reduced after only a month? What could be the mechanism? What is the blood pressure in these mice? I assume that brain endothelial cells are also targeted and thus, the brain/heart axis must be affected also.

We thank this reviewer for this comment. In the revised manuscript, we performed the immunohistochemical staining of cardiac structure and collagen. The collagen content was increased in AAV-PHGDH and AAV-PKM2 mice (**Supplementary Fig. 9b**), consistent with improved cardiac functions in AAV-PHGDH and AAV-PKM2 mice.

As for the heart rate, we think it may be due to the number of mice tested (n=6) is small. In the revised manuscript, we performed the in vivo experiment with more mice (n=23, 12 male, 11 female). We did not observe significantly reduced heart rate in AAV-PHGDH and AAV-PKM2 mice (**Supplementary Fig. 9e**). We also measured arterial blood pressure. AAV-PHGDH and AAV-PKM2 mice had no significantly changed systolic pressure and diastolic pressure compared with AAV-Ctrl (**Supplementary Fig. 9c, d**).

As for the expression of PHGDH and PKM2 in the brain, we detected the little but detectable expression of Myc-PHGDH and FLAG-PKM2 in the brain samples from AAV-PHGDH and AAV-PKM2 mice (**Supplementary Fig. 8e**), suggesting that the brain endothelial cells are mildly affected.

Page 20, paragraph 1: Added “We also examined the expression of AAV-PHGDH and AAV-PKM2 in the brain and trace amount was detected (Supplementary Fig. 8e), suggesting that the brain endothelial cells are mildly affected.”

Page 21, paragraph 1: Added “VE-specific expression of PHGDH and PKM2 also increased collagen content in the heart (Supplementary Fig. 9b). We then measured the effect of VE-targeted expression of PHGDH and PKM2 on arterial blood pressure. AAV-PHGDH and AAV-PKM2 mice had similar systolic pressure and diastolic pressure as AAV-Ctrl (Supplementary Fig. 9c, d).”

4. Finally, a true impact on endothelial function would be ideally to demonstrate a better endothelial flow-mediated dilation, or at least endothelium-dependent relaxation *ex vivo* in arteries.

We agree with the reviewer’s comment. In the revised manuscript, we isolated the aortae from AAV-Ctrl, AAV-PHGDH and AAV-PKM2 mice and measured the wall tension after *ex vivo* mechanical stretch according to protocols described by Voelkl et al (*J Clin Invest* 2018, 128:3024-3040). The aortae from AAV-PHGDH and AAV-PKM2 mice developed less wall tension after *ex vivo* mechanical stretch as compared with AAV-Ctrl (**Fig. 7j**), indicative of less stiffness in AAV-PHGDH and AAV-PKM2 mice.

Page 21, paragraph 1: Added “The aortae from AAV-PHGDH and AAV-PKM2 mice developed less wall tension after *ex vivo* mechanical stretch as compared with AAV-Ctrl (Fig. 7j), indicative of less stiffness in AAV-PHGDH and AAV-PKM2 mice.”

5. The *in vivo* data are impressive and fascinating, but they need a direct validation that the effects seen are solely due to a reduction in the number of senescent endothelial cells as claimed; in addition, there is no evidence *in vivo* from the aorta data whether the transfection slowed senescence or had a senolytic effect (killed senescent cells).

To directly validate the effects seen are due to a reduction in the number of senescent cells, we performed SA- β -gal staining of the aortae from AAV-Ctrl, AAV-PHGDH and AAV-PKM2 mice. For the aortae from AAV-PHGDH and AAV-PKM2 mice, a significant decrease in SA- β -gal staining signal was observed when compared to that of AAV-Ctrl mice (**Supplementary Fig. 9j**). We also examined the senolytic effect caused by over-expression of PKM2 and PHGDH. By examining the expression of pro-caspase 3 and cleaved caspase 3, we found that over-expression of PHGDH and PKM2 had no significant effect on the expression of pro-caspase 3 and cleaved caspase 3 (**Fig. 7g; Supplementary Fig. 8d**). Although our effort to performed SA- β -gal staining for the endothelium was unsuccessful, these data still suggest that the AAV-PHGDH and AAV-PKM2 transfection attenuates cell senescence and improves cardiac function.

Page 22, paragraph 2: Added “To examine the above effects were caused by reduction of senescent endothelial cells or senolytic effects, we first performed SA- β -gal staining of the aorta from AAV-Ctrl, AAV-PHGDH and AAV-PKM2 mice. Overexpression of PHGDH and PKM2 significantly decreased SA- β -gal staining signal (Supplementary

Fig. 9i). We then examined the senolytic effect from over-expression of PKM2 and PHGDH. The expression of pro-apoptotic protein (pro-caspase 3, cleaved caspase 3) was unchanged among the aortae from AAV-Ctrl, AAV-PHGDH and AAV-PKM2 mice (Fig. 7g; Supplementary Fig. 8d). All these data suggest that the transfection slowed cell senescence with little senolytic effect.”

6. Sex is not considered and it should be discussed; it could be a significant parameter considering that E2 is protective and may delay senescence of endothelial cells in females.

That is a good point. It is true that estrogen is protective for cardiovascular function and estrogen is lost with aging (*Pharmacol Ther* 2012, 135(1): 54-70). Estrogen has been reported to reduce endothelial progenitor cell (EPC) senescence through augmentation of telomerase activity (*Ther Adv Cardiovasc Dis* 2010, 4(1):55-69). It is possible that estrogen may delay the senescence of endothelial cells in females. In the revised manuscript, we re-performed the in vivo experiments and compared the effect of AAV-PHGDH and AAV-PKM2 on male and female mice. There was no marked difference for male and female mice on heart weight, heart rate for male and female (**Fig. 7h, i**). But for LV ejection fraction, LV fractional shortening and exercise capability, overexpression of PHGDH and PKM2 displayed a little better beneficial effect on female mice than male mice (**Fig. 7k-m; Supplementary Fig. 9f**). This better effects in female mice may be related to the role of estrogen in delaying senescence. We discussed the gender difference in the discussion section.

Page 28, paragraph 2: Added “Estrogen is protective for cardiovascular function and estrogen is lost with aging ⁶⁰. Estrogen has been reported to reduce endothelial progenitor cell (EPC) senescence through augmentation of telomerase activity ⁶¹. It is possible that estrogen may delay the senescence of endothelial cells in females. We compared the gender difference in our in vivo experiments. There was no marked difference for male and female mice on heart weight, heart rate for male and female (Fig. 7h, i). For LV ejection fraction, LV fractional shortening and exercise capability, overexpression of PHGDH and PKM2 displayed a little better beneficial effect in female mice than male mice (Fig. 7k-m; Supplementary Fig. 9f). This better effect in female mice may be related to the role of estrogen in delaying senescence.”

7. In any case, the overall data are impressive and the potential impact of the data in the field are high.

We thank this reviewer for the recognition of our work and those helpful comments, which have significantly improved the quality of our manuscript.

Reviewer #4 (Remarks to the Author):

In this paper, the authors suggested that serine is essential for attenuating the aging of endothelial cells. Although the role of serine in aging has been implicated previously, they revealed pyruvate kinase M2 (PKM2) as a target for serine and revealed that PKM2 interacts with phosphoglycerate dehydrogenase (PHGDH), a serine biosynthetic rate-limiting enzyme. They also elucidated the mechanism by which PHGDH regulates senescence-associated genes such as SIRT1 by increasing H3T11 phosphorylation through PKM2-K433 acetylation and nuclear translocation. The authors performed well-designed in vitro experiments to verify PHGDH-PKM2-H3pT11 axis and confirmed that PHGDH and PKM2 are involved in ameliorating the aging phenotype in vivo. However, following concerns should be addressed before final decision to publication.

We sincerely thank this reviewer for recognition of our work. We also thank this reviewer for the constructive suggestions and comments, which have significantly improved the quality of this manuscript. We have revised the manuscript according to the suggestions as requested.

Major Points

1. Previously, the relationship between serine and aging has already been indicated; through the studies that endogenous d-serine levels in the hippocampus decrease with aging [1] and that serine induces life span extension [2] etc. Therefore, it is better to explain the significance and novelty of this study compared to previous studies.

[1] Potier B, et al. (2010). *Front Aging Neurosci.* 2:1. [2] Edwards C, et al. (2015). *BMC Genet*, 16:8.

We thank this reviewer for pointing out these two papers. It has been reported that D-serine but not L-serine alleviates the age-related deficit in synaptic plasticity (*Front Aging Neurosci* 2010, 2:1). However, D-serine was predominantly found in the brain tissues and its amount was very low in other organs, such as liver, kidney, serum and spinal cord (*J Neurochem* 1993, 60(2):783-6). In contrast, L-serine ubiquitously exists in these organs (*J Neurochem* 1993, 60(2):783-6). It is possible that L-serine but not D-serine has a more important role in regulating the aging of these organs. Moreover, L-serine but not D-serine supplementation has been reported to cause lifespan extension in *C. elegans* (*BMC Genet* 2015, 16:8). This lifespan extension appears to be caused by altered mitochondrial metabolism and activation of stress response pathways, including DAF-16/FOXO and SKN-1/Nrf2 (*BMC Genet* 2015, 16:8). It remains unknown whether it is L-serine or its metabolism mediates this effect. Here, we report that L-serine per se but not its metabolism attenuates senescence in endothelial cells by acting as a signal molecule to activate PKM2. Moreover, we identify a novel function of serine biosynthetic enzyme PHGDH in regulating cellular senescence. We have added these in the discussion section.

Page 24, paragraph 2: Added “It has been reported that D-serine but not L-serine alleviates the age-related deficit in synaptic plasticity ⁵². However, D-serine was predominantly found in the brain tissues and its amount was very low in other organs, such as liver, kidney, serum and spinal cord ⁵³. In contrast, L-serine ubiquitously exists in these organs ⁵³. It is possible that L-serine but not D-serine has a more important role in regulating the aging of these organs. Moreover, L-serine but not D-serine supplementation has been reported to cause lifespan extension in *C. elegans* ⁵⁴. This lifespan extension appears to be caused by altered mitochondrial metabolism and activation of stress response pathways, including DAF-16/FOXO and SKN-1/Nrf2 ⁵⁴. It remains unknown whether it is L-serine or its metabolism mediates this effect. Here, we report that L-serine per se but not its metabolism attenuates senescence in endothelial cells by acting as a signal molecule to activate PKM2. Moreover, we identify a novel function of serine biosynthetic enzyme PHGDH in regulating cellular senescence.”

2. The authors used DARTS-LC-MS/MS method to investigate targets that resist degradation by pronase treatment due to binding of ligand small molecule, serine. However, there is no rationale for choosing 'PKM2' as the target of serine. Did the sequence coverage value of PKM2 belong to the top? Please provide in detail the criteria how target candidates were selected from detected proteins and provide the list of target candidates with high sequence coverage value.

We thank this reviewer for this comment. We used the enrichment to score the proteins in the DARTS-LC-MS data. The enrichment score was calculated by dividing the spectra of serine-protected samples to the spectra of control samples. In addition, we also added the following criteria: the number of peptides in control samples is less than 20; the number of peptides in serine-protected are more than 20. The target candidates were ranked according to their scores (**Supplementary Fig. 3a**). We added the list of target candidates and the score in Supplementary Fig. 3a.

We added this information in the Figure legend of Supplementary Fig. 3a: “List of potential proteins that are bound by serine as determined by DARTS-MS. The enrichment was calculated by dividing the spectra of serine-protected samples to the spectra of control samples. In addition, the following criteria was used: the number of peptides in control samples is less than 20; the number of peptides in serine-protected are more than 20. The target candidates were ranked according to their scores.”

3. In Fig 2b, the authors performed DARTS after expressing WT PKM2 or PKM2 H464A in cells and showed that PKM2 H464A mutant did not bind to serine supporting PKM2 could be a serine binding protein. However, the PKM2 antibody cannot differentiate between endogenous PKM2 and expressed WT PKM2, PKM2 H464A. If

the expression vector includes the tag (FLAG or Myc?), it would be better to observe the DARTS pattern for ectopic PKM2 using FLAG or Myc antibodies instead. In addition, it would be better to investigate the binding activity of PKM2 H464A (serine non-binding mutant) with PHGDH and its phosphorylation activity to H3T11 in the regulation of senescence-associated genes (*SIRT1*, *BMI1*, *LAMA1*) regulation.

We thank this reviewer for this comment. In Fig. 2b, we indeed used anti-FLAG antibody to detect the ectopically expressed FLAG-PKM2 (**Fig. 2b**). The endogenous PKM2 cannot be detected with this antibody. To make this clearer, we added WT FLAG-PKM2 and FLAG-PKM2 H464A labels in Fig. 2b. Meanwhile, we added the antibody information in the figure legend.

We performed Co-IP to examine the binding activity of PKM2 H464A with PHGDH. Our data showed that PKM2 H464A had similar binding ability to PHGDH as WT PKM2 (**Supplementary Fig. 4d**). In addition, we examined the effect of PKM2-H464A on H3T11 phosphorylation at senescence-associated genes (*SIRT1*, *BMI1*, *LAMA1*). Exogenous serine increased H3pT11 at the promoters of these three genes in WT PKM2 cells but not in PKM2-H464A mutant (**Supplementary Fig. 7o**), which is consistent with our immunoblot data (**Supplementary Fig. 5i**). In accordance with H3pT11 changes, exogenous serine induced the expression of these three genes in WT PKM2 cells but not in PKM2-H464A mutant (**Supplementary Fig. 7n**). These data further support our conclusion that the PHGDH-PKM2-H3pT11 axis regulates the transcription of senescence-associated genes in ECs. We have put these data in the revised manuscript.

Page 10, paragraph 1: Added “Mutation of PKM2 H464A had no effect on the interaction between PKM2 and PHGDH (Supplementary Fig. 4d).”

Page 18, paragraph 2: Added “Mutation of PKM2 H464A also abolished serine-induced the expression of *SIRT1*, *BMI1* and *LAMA1* (Supplementary Fig. 7n)”.

Page 18, paragraph 2: Added “Exogenous serine increased the enrichment of H3pT11 at *SIRT1*, *BMI1* and *LAMA1* in HUVECs that overexpress WT PKM2 but not PKM2 H464A mutant (Supplementary Fig. 7o)”.

Minor points

1. Add information about the number of passages of HUVECs, pronase and serine concentration, pronase treated time for DARTS experiments.

We added the detailed information in the figure legend.

Page 47, paragraph 2: Changed “HeLa lysates were incubated with or without serine and then subjected to pronase digestion” to “HeLa lysates were incubated with or

without 300 μ M serine and then subjected to pronase (50 μ g/ml) digestion for 15 min”.

Page 47, paragraph 2: Changed “The lysates of HUVECs that stably overexpress WT PKM2 and PKM2 H464A were incubated with serine and then subjected to pronase digestion.” to “The lysates of HUVECs (P10) that stably overexpress WT PKM2 and PKM2 H464A were incubated with 0-600 μ M serine and then subjected to pronase (50 μ g/ml) digestion for 15 min.”

2. [Figure 2a] In CBB-stained gel images, it is difficult to determine whether the indicated band is exactly PKM2 or other proteins of similar size.

We thank this reviewer for this comment. We removed the label for PKM2 in Fig. 2a.

3. [Figure 2b] The authors used two serine concentrations, and no concentration dependence was observed. Please add possible discussion for this.

We thank this reviewer for this comment. The possible reason could be due to the concentration of serine used for DARTS is too high to observe the concentration-dependent interaction. In the revised manuscript, we reduced the serine concentration to 300 μ M and 600 μ M in DARTS experiments. We observed the concentration-dependent increase of serine binding to PKM2 (**Fig. 2b**).

Page 47, paragraph 2: Changed “The lysates of HUVECs that stably overexpress WT PKM2 and PKM2 H464A were incubated with serine and then subjected to pronase digestion.” to “The lysates of HUVECs (P10) that stably overexpress WT PKM2 and PKM2 H464A were incubated with 0-600 μ M serine and then subjected to pronase (50 μ g/ml) digestion for 15 min.”

REVIEWER COMMENTS

Reviewer #1 (Remarks to the Author):

The authors have properly answered my concerns and the manuscript has been improved after revision.

Reviewer #2 (Remarks to the Author):

Authors have addressed my comments and critiques. No further concerns.

Reviewer #3 (Remarks to the Author):

The authors have addressed some of the previous concerns but some issues still need to be resolved.

First, a comment: it is concerning that the cardiac phenotype (reduced HR) is no longer seen after doubling the number of animals in the group from 6 to 12 in male mice and overall 23 by including female mice. But I admit that functional studies can be variable.

Suppl Fig 9b: collagen. First, it needs to be quantified if it is to be interpreted. Second, usually, if it is increased, cardiac function decreases because of the stiffening of the ventricle. Usually, collagen accumulation is associated with fibrosis.

Suppl Fig 9j: b-Gal needs to be quantified to be used in the discussion.

Endothelial function. The data presented in figure 7j are not representative of the endothelial function as requested. The data show that the arterial wall is less stiff. The method used from Voelkl et al (J Clin Invest 2018, 128:3024-3040) was indeed to assess the relationship between vascular calcification and stiffness. It is an interesting phenotype because age-associated stiffness is considered as deleterious by promoting systolic hypertension and end-organ damage. However, the occurrence of the "de-stiffening" is fast and should be documented by assessment of the structure of the wall because it has not been seen before in old mice.

To assess endothelial function, as suggested, arterial rings mounted in the organ bath on the force transducers, but rather than stretching the rings, a precontraction could have been induced by phenylephrine and, at equilibrium, an endothelium-dependent relaxation induced by the addition of increasing concentrations of acetylcholine.

Line 631. Replace "gender" by "sex". Gender is not assessed in mice.

Reviewer #4 (Remarks to the Author):

The manuscript is well complemented, and the authors responded properly to the concerns of this reviewer to improve the readability of the manuscript. The authors newly presented the importance of L-serine, not D-serine, which was mainly focused in the previous studies. However, since this indication is described for the first time in the discussion part, this reviewer recommends that 'serine' should be highlighted as 'L-serine (hereinafter referred to as serine)' in the early part of the manuscript.

Point-by-point response to the referees' comments

REVIEWER COMMENTS

Our responses are in blue.

Reviewer #1 (Remarks to the Author):

The authors have properly answered my concerns and the manuscript has been improved after revision.

Reviewer #2 (Remarks to the Author):

Authors have addressed my comments and critiques. No further concerns.

Reviewer #3 (Remarks to the Author):

The authors have addressed some of the previous concerns but some issues still need to be resolved.

First, a comment: it is concerning that the cardiac phenotype (reduced HR) is no longer seen after doubling the number of animals in the group from 6 to 12 in male mice and overall 23 by including female mice. But I admit that functional studies can be variable.

We would like to thank this reviewer for this comment. We agree with the reviewer that functional studies may be variable especially for those with mild effect. Sometimes sufficient number of animals may be required to obtain reliable results. Nonetheless, we are sure that the effect of over-expression of PHGDH and PKM2 on endothelial and cardiac functions is reproducible.

Suppl Fig 9b: collagen. First, it needs to be quantified if it is to be interpreted. Second, usually, if it is increased, cardiac function decreases because of the stiffening of the ventricle. Usually, collagen accumulation is associated with fibrosis.

We would like to thank this reviewer for this point. We quantified the collagen contents and put the data in Supplementary Fig. 9b. Our data showed that over-expression of PKM2 and PHGDH significantly increased the collagen content. This could be a side effect of PKM2 on biosynthesis of glycine, which has been reported to increase the synthesis of collagen (*Theranostics* 2021, 11(19):9331-9341). Myofibroblast differentiation upregulates PKM2 and promotes its dimerization. PKM2 dimer slows the flow rate of glycolysis and channels glycolytic intermediates to de novo glycine synthesis, which facilitates collagen synthesis and secretion in myofibroblasts (*Theranostics* 2021, 11(19):9331-9341). Thus, PKM2 dimer not only acts as a protein kinase to phosphorylate H3T11 but also promotes collagen synthesis.

Several studies showed that the aorta stiffness is not simply determined by the content of collagen but the composition of collagen subtypes. For example, the degradation of collagen type-I is increased in subjects with stiffer arteries (*J Hum Hypertens* 2006, 20(11):867-73). In spontaneously hypertensive rats, there were decreased amounts of collagen type-I accompanied by an increase in collagen type-V (*J Lab Clin Med* 1989, 113(5):604-11). This change in the composition of collagen subtypes was associated with increased aortic stiffness (*J Lab Clin Med* 1989, 113(5):604-11). Hence, whether over-expression of PKM2 and PHGDH affects aortic stiffness and fibrosis depends on their effect on the composition of collagen subtypes not just the amount of collagen. Our data showed that over-expression of PKM2 and PHGDH attenuates cellular senescence and improves endothelial and cardiac functions, suggesting that the beneficial effect of PKM2 and PHGDH exceeds the side effect. Nevertheless, we admit that it may need to consider how to prevent the potential side effect of over-expression of PKM2 and PHGDH on glycine synthesis. One possible way to reduce glycine synthesis is blocking the activity of SHMT1/2, which are the enzymes that convert serine to glycine. In the revised manuscript, we added this information as the limitations of this study in the discussion section.

Line 469: Changed “VE-specific expression of PHGDH and PKM2 increased collagen content in the heart (Supplementary Fig. 9b)” to “VE-specific expression of PHGDH and PKM2 increased collagen content in the heart (Supplementary Fig. 9b), which is consistent with the reports that PKM2 dimer promotes glycine biosynthesis to facilitate collagen accumulation in myofibroblasts⁴⁵”.

Line 646: Added “Lastly, our study showed that over-expression of PKM2 and PHGDH increased the amount of collagen III, which may facilitate fibrosis. When PHGDH and PKM2 are needed to be over-expressed, it is better to prevent their potential side effect on glycine biosynthesis. One possible way is blocking the activity of SHMT1/2, which are the enzymes that convert serine to glycine.”

Suppl Fig 9j: b-Gal needs to be quantified to be used in the discussion.

We quantified the β -Gal staining in Supplementary Fig. 9j as requested.

Endothelial function. The data presented in figure 7j are not representative of the endothelial function as requested. The data show that the arterial wall is less stiff. The method used from Voelkl et al (*J Clin Invest* 2018, 128:3024-3040) was indeed to assess the relationship between vascular calcification and stiffness. It is an interesting phenotype because age-associated stiffness is considered as deleterious by promoting systolic hypertension and end-organ damage. However, the occurrence of the "de-stiffening" is fast and should be documented by assessment of the structure of the wall

because it has not been seen before in old mice.

To assess endothelial function, as suggested, arterial rings mounted in the organ bath on the force transducers, but rather than stretching the rings, a precontraction could have been induced by phenylephrine and, at equilibrium, an endothelium-dependent relaxation induced by the addition of increasing concentrations of acetylcholine.

We thank this reviewer for this comment. We performed the experiments to assess the endothelial functions as requested. The arterial rings were mounted in the organ bath of a force transducers, pre-contracted with phenylephrine and at equilibrium, the relaxation was measured after induction with increasing concentrations of acetylcholine. Our data showed that VE-targeted expression of PKM2 and PHGDH in aged mice significantly increased the relaxation of arterial rings (Fig. 7j). As the reviewer thought the wall tension assay in original Fig. 7 is not representative of the endothelial function, we replaced the wall tension data with the arterial rings relaxation results in the revised manuscript.

Line 466: Changed “The aortae from AAV-PHGDH and AAV-PKM2 mice developed less wall tension after ex vivo mechanical stretch as compared with AAV-Ctrl (Fig. 7j), indicative of less stiffness in AAV-PHGDH and AAV-PKM2 mice” to “To assess the effect of VE-targeted expression of PHGDH and PKM2 on endothelial functions, we first isolated the aortae from AAV-PHGDH and AAV-PKM2 mice and examined their response to acetylcholine. VE-targeted expression of PHGDH and PKM2 significantly increased endothelium-dependent relaxation when compared with AAV-Ctrl (Fig. 7j)”.

Line 631. Replace "gender" by "sex". Gender is not assessed in mice.

We changed “gender” to “sex” as requested.

Line 634: Changed “We compared the gender difference in our in vivo experiments” to “We compared the sex difference in our in vivo experiments”.

Reviewer #4 (Remarks to the Author):

The manuscript is well complemented, and the authors responded properly to the concerns of this reviewer to improve the readability of the manuscript. The authors newly presented the importance of L-serine, not D-serine, which was mainly focused in the previous studies. However, since this indication is described for the first time in the discussion part, this reviewer recommends that ‘serine’ should be highlighted as ‘L-serine (hereinafter referred to as serine)’ in the early part of the manuscript.

We would like to thank this reviewer for this comment. We changed “serine” to “L-serine (hereinafter referred to as serine)”.

Line 50: Changed “Recent study showed that glycolysis-derived serine biosynthesis” to “Recent study showed that glycolysis-derived L-serine (hereinafter referred to as serine) biosynthesis’.

REVIEWERS' COMMENTS

Reviewer #3 (Remarks to the Author):

The authors have appropriately responded to my comments. I commend them for this work.